# ETSFORMER: EXPONENTIAL SMOOTHING TRANSFORMERS FOR TIME-SERIES FORECASTING

## ABSTRACT

Transformers have recently been actively studied for time-series forecasting. While often showing promising results in various scenarios, traditional Transformers are not designed to fully exploit the characteristics of time-series data and thus suffer some fundamental limitations, e.g., they are generally not decomposable or interpretable, and are neither effective nor efficient for long-term forecasting. In this paper, we propose ETSformer, a novel time-series Transformer architecture, which exploits the principle of exponential smoothing methods in improving Transformers for time-series forecasting. Specifically, ETSformer leverages a novel level-growth-seasonality decomposed Transformer architecture which leads to more interpretable and disentangled decomposed forecasts. We further propose two novel attention mechanisms – the exponential smoothing attention and frequency attention, which are specially designed to overcome the limitations of the vanilla attention mechanism for time-series data. Extensive experiments on the long sequence time-series forecasting (LSTF) benchmark validates the efficacy and advantages of the proposed method. Code is attached in the supplementary material, and will be made publicly available.

## 1 INTRODUCTION

Transformer models have achieved great success in the fields of natural language processing (Vaswani et al., 2017; Devlin et al., 2019), computer vision (Carion et al., 2020; Dosovitskiy et al., 2021), and even more recently, time-series (Li et al., 2019; Wu et al., 2021; Zhou et al., 2021; Zerveas et al., 2021; Zhou et al., 2022). While the success of Transformer models have been widely attributed to the self-attention mechanism, alternative forms of attention, infused with the appropriate inductive biases, have been introduced to tackle the unique properties of their underlying task or data (You et al., 2020; Raganato et al., 2020). In time-series forecasting, decomposition-based architectures such as Autoformer and FEDformer models (Wu et al., 2021; Zhou et al., 2022) have incorporated time-series specific inductive biases, leading to increased accuracy, and more interpretable forecasts (by decomposing forecasts into seasonal and trend components). Their success has been motivated by: (i) disentangling seasonal and trend representations via seasonal-trend decomposition (Cleveland & Tiao, 1976; Cleveland et al., 1990; Woo et al., 2022), and (ii) replacing the vanilla pointwise dot-product attention which handle time-series patterns such as seasonality and trend inefficiently, with time-series specific attention mechanisms such as the Auto-Correlation mechanism and Frequency-Enhanced Attention. While these existing work introduce the promising direction of interpretable and decomposed time-series forecasting for Transformer-based architectures, they suffer from two drawbacks.

Firstly, they suffer from entangled seasonal-trend representations, evidenced in Figure 1, where the trend forecasts exhibit periodical patterns which should only be present in the seasonal component, and the seasonal component does not accurately track the (multiple) periodicities present in the ground truth seasonal component. This arises due to their decomposition mechanism which detects trend via a simple moving average over the input signal and detrends the signal by removing the detected trend component – an arguably naive approach. This method has many known pitfalls (Hyndman & Athanasopoulos, 2018), such as the trend-cycle component not being available for the first and last few observations, and over-smoothing rapid rises and falls.

Secondly, their proposed replacements for the vanilla attention mechanism are not human interpretable – demonstrated in Section 3.3. Model inspection and diagnosis allows us to better understand the fore-

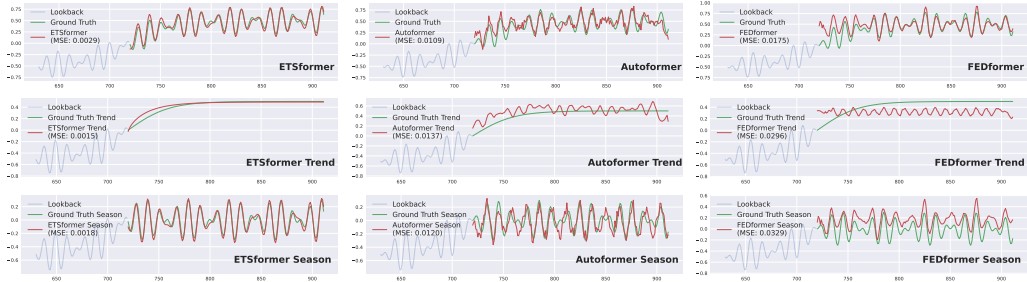

Figure 1: Seasonal-trend decomposed forecasts on synthetic data with ground truth seasonal and trend components. Top row: combined forecast. Middle row: trend component forecast. Bottom row: season component forecast. ETSformer is compared to two competing decomposed Transformer baselines, Autoformer, and FEDformer. Seen in the visualization, ETSformer exhibits a more disentangled seasonal-trend decomposition which accurately tracks the ground truth components. Not visualized here is ETSformer's unique ability to further separate trend into level and growth components.

casts generated by our models, attributing predictions to each component to make better downstream decisions. For an attention mechanism focusing on seasonality, we would expect the cross-attention map visualization to produce clear periodic patterns which shift smoothly across decoder time steps. Yet, the Auto-Correlation mechanism from Autoformer does not exhibit this property, yielding similar attention weights across decoder time steps, while the Frequency-Enhanced Attention from FEDformer does not have such model interpretability capabilities due to its complicated frequency domain attention.

To address these limitations, we look towards the more principled approach of level-growth-season decomposition from ETS methods (Hyndman et al., 2008) (further introduced in Appendix A). This principle further deconstructs trend into level and growth components. To extract the level and growth components, we also look at the idea of exponential smoothing, where more recent data gets weighted more highly than older data, reflecting the view that the more recent past should be considered more relevant for making new predictions or identifying current trends, to replace the naive moving average. At the same time, we leverage the idea of extracting the most salient periodic components in the frequency domain via the Fourier transform, to extract the global seasonal patterns present in the signal. These principles help yield a stronger decomposition strategy by first extracting global periodic patterns as seasonality, and subsequently extracting growth as the change in level in an exponentially smoothed manner.

Motivated by the above, we propose ETSformer, an interpretable and efficient Transformer architecture for time-series forecasting which yields disentangled seasonal-trend forecasts. Instead of reusing the moving average operation for detrending, ETSformer overhauls the existing decomposition architecture by leveraging the level-growth-season principle, embedding it into a novel Transformer framework in a non-trivial manner. Next, we introduce interpretable and efficient attention mechanisms – Exponential Smoothing Attention (ESA) for trend, and Frequency Attention (FA) for seasonality. ESA assigns attention weights in an exponentially decreasing manner, with high values to nearby time steps and low values to far away time steps, thus specialising in extracting growth representations. FA leverages frequency domain representations to extract dominating seasonal patterns by selecting the Fourier bases with the $K$ largest amplitudes. Both mechanisms have efficient implementations with $\mathcal{O}(L \log L)$ complexity. Furthermore, we demonstrate human interpretable visualizations of both mechanisms in Section 3.3. To summarize, our key contributions are as follows:

- We introduce a novel decomposition Transformer architecture, incorporating the time-tested level-growth-season principle for more disentangled, human-interpretable time-series forecasts.

- We introduce two new attention mechanisms, ESA and FA, which incorporate stronger time-series specific inductive biases. They achieve better efficiency than vanilla attention, and yield interpretable attention weights upon model inspection.

- The resulting method is a highly effective, efficient, and interpretable deep forecasting model. We show this via extensive empirical analysis, that ETSformer achieves performance competitive with state-of-the-art methods over 6 real world datasets on both multivariate and univariate settings, and is highly efficient compared to competing methods.

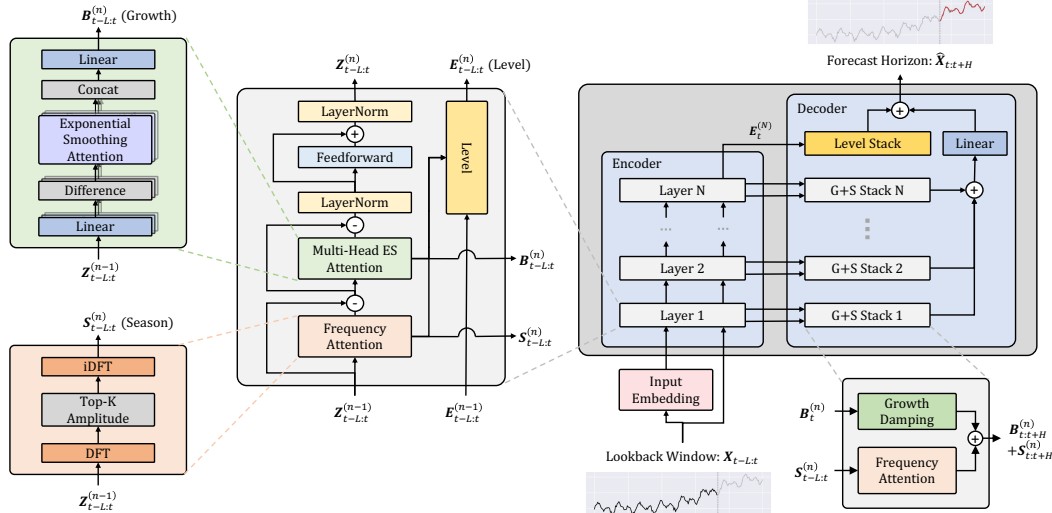

Figure 2: ETSformer model architecture.

## 2 ETSFORMER

**Problem Formulation**  Let $\boldsymbol{x}_t \in \mathbb{R}^m$ denote an observation of a multivariate time-series at time step $t$. Given a lookback window $\boldsymbol{X}_{t-L:t} = [\boldsymbol{x}_{t-L}, \ldots, \boldsymbol{x}_{t-1}]$, we consider the task of predicting future values over a horizon, $\boldsymbol{X}_{t:t+H} = [\boldsymbol{x}_t, \ldots, \boldsymbol{x}_{t+H-1}]$. We denote $\hat{\boldsymbol{X}}_{t:t+H}$ as the point forecast of $\boldsymbol{X}_{t:t+H}$. Thus, the goal is to learn a forecasting function $\hat{\boldsymbol{X}}_{t:t+H} = f(\boldsymbol{X}_{t-L:t})$ by minimizing some loss function $\mathcal{L} : \mathbb{R}^{H \times m} \times \mathbb{R}^{H \times m} \to \mathbb{R}$.

In the following, we explain how ETSformer infuses level-growth-seasonal decomposition the the classical encoder-decoder Transformer architecture, specializing for interpretable time-series forecasting. Our architecture design methodology relies on three key principles: (1) the architecture leverages the stacking of multiple layers to progressively extract a series of level, growth, and seasonal representations from the intermediate latent residual; (2) performs level-growth-seasonal decomposition of latent representations, by extracting salient seasonal patterns while modeling level and growth components following an exponential smoothing formulation; (3) the final forecast is a composition of level, growth, and seasonal components making it human interpretable.

### 2.1 OVERALL ARCHITECTURE

Figure 2 illustrates the overall encoder-decoder architecture of ETSformer. At each layer, the encoder is designed to iteratively extract growth and seasonal latent components from the lookback window. The level is then extracted in a similar fashion to classical level smoothing in Equation (3). These extracted components are then fed to the decoder to further generate the final $H$-step ahead forecast via a composition of level, growth, and seasonal forecasts, which is defined:

$$\hat{\boldsymbol{X}}_{t:t+H} = \boldsymbol{E}_{t:t+H} + \text{Linear}\Big( \sum_{n=1}^{N} (\boldsymbol{B}_{t:t+H}^{(n)} + \boldsymbol{S}_{t:t+H}^{(n)}) \Big), \tag{1}$$

where $\boldsymbol{E}_{t:t+H} \in \mathbb{R}^{H \times m}$, and $\boldsymbol{B}_{t:t+H}^{(n)}, \boldsymbol{S}_{t:t+H}^{(n)} \in \mathbb{R}^{H \times d}$ represent the level forecasts, and the growth and seasonal latent representations of each time step in the forecast horizon, respectively. The superscript represents the stack index, for a total of $N$ encoder stacks. Note that $\text{Linear}(\cdot) : \mathbb{R}^d \to \mathbb{R}^m$ operates element-wise along each time step, projecting the extracted growth and seasonal representations from latent to observation space.

#### 2.1.1 INPUT EMBEDDING

Raw signals from the lookback window are mapped to latent space via the input embedding module, defined by $\boldsymbol{Z}_{t-L:t}^{(0)} = \boldsymbol{E}_{t-L:t}^{(0)} = \text{Conv}(\boldsymbol{X}_{t-L:t})$, where Conv is a temporal convolutional filter with kernel size 3, input channel $m$ and output channel $d$. In contrast to prior work (Li et al., 2019; Wu et al., 2020; 2021; Zhou et al., 2021), the inputs of ETSformer do not rely on any other manually

designed dynamic time-dependent covariates (e.g. month-of-year, day-of-week) for both the lookback window and forecast horizon. This is because the proposed Frequency Attention module (details in Section 2.2.2) is able to automatically uncover these seasonal patterns, which renders it more applicable for challenging scenarios without these discriminative covariates and reduces the need for feature engineering.

### 2.1.2 ENCODER

The encoder focuses on extracting a series of latent growth and seasonality representations in a cascaded manner from the lookback window. To achieve this goal, traditional methods rely on the assumption of additive or multiplicative seasonality which has limited capability to express complex patterns beyond these assumptions. Inspired by (Oreshkin et al., 2019; He et al., 2016), we leverage residual learning to build an expressive, deep architecture to characterize the complex intrinsic patterns. Each layer can be interpreted as sequentially analyzing the input signals. The extracted growth and seasonal signals are then removed from the residual and undergo a nonlinear transformation before moving to the next layer. Each encoder layer takes as input the residual from the previous encoder layer $\boldsymbol{Z}_{t-L:t}^{(n-1)}$ and emits $\boldsymbol{Z}_{t-L:t}^{(n)}, \boldsymbol{B}_{t-L:t}^{(n)}, \boldsymbol{S}_{t-L:t}^{(n)}$, the residual, latent growth, and seasonal representations for the lookback window via the Multi-Head Exponential Smoothing Attention (MH-ESA) and Frequency Attention (FA) modules (detailed description in Section 2.2). The following equations formalizes the overall pipeline in each encoder layer, and for ease of exposition, we use the notation $:=$ for a variable update.

$$\text{Seasonal:} \quad \boldsymbol{S}_{t-L:t}^{(n)} = \text{FA}_{t-L:t}(\boldsymbol{Z}_{t-L:t}^{(n-1)}) \qquad \text{Growth:} \quad \boldsymbol{B}_{t-L:t}^{(n)} = \text{MH-ESA}(\boldsymbol{Z}_{t-L:t}^{(n-1)})$$

$$\boldsymbol{Z}_{t-L:t}^{(n-1)} := \boldsymbol{Z}_{t-L:t}^{(n-1)} - \boldsymbol{S}_{t-L:t}^{(n)} \qquad\qquad \boldsymbol{Z}_{t-L:t}^{(n-1)} := \text{LN}(\boldsymbol{Z}_{t-L:t}^{(n-1)} - \boldsymbol{B}_{t-L:t}^{(n)})$$

$$\boldsymbol{Z}_{t-L:t}^{(n)} = \text{LN}(\boldsymbol{Z}_{t-L:t}^{(n-1)} + \text{FF}(\boldsymbol{Z}_{t-L:t}^{(n-1)}))$$

LN is layer normalization (Ba et al., 2016), $\text{FF}(x) = \text{Linear}(\sigma(\text{Linear}(x)))$ is a position-wise feedforward network (Vaswani et al., 2017) and $\sigma(\cdot)$ is the sigmoid function.

**Level Module** Given the latent growth and seasonal representations from each layer, we extract the level at each time step $t$ in the lookback window in a similar way as the level smoothing equation in Equation (3). Formally, the adjusted level is a weighted average of the current (de-seasonalized) level and the level-growth forecast from the previous time step $t - 1$. It can be formulated as:

$$\boldsymbol{E}_t^{(n)} = \boldsymbol{\alpha} * \left( \boldsymbol{E}_t^{(n-1)} - \text{Linear}(\boldsymbol{S}_t^{(n)}) \right) + (1 - \boldsymbol{\alpha}) * \left( \boldsymbol{E}_{t-1}^{(n)} + \text{Linear}(\boldsymbol{B}_{t-1}^{(n)}) \right),$$

where $\boldsymbol{\alpha} \in \mathbb{R}^m$ is a learnable smoothing parameter, $*$ is an element-wise multiplication term, and $\text{Linear}(\cdot) : \mathbb{R}^d \to \mathbb{R}^m$ maps representations to observation space. Finally, the extracted level in the last layer $\boldsymbol{E}_{t-L:t}^{(N)}$ can be regarded as the corresponding level for the lookback window. We show in Appendix B.3 that this recurrent exponential smoothing equation can also be efficiently evaluated using the efficient $\mathcal{A}_{\text{ES}}$ algorithm (Algorithm 1) with an auxiliary term.

### 2.1.3 DECODER

The decoder is tasked with generating the $H$-step ahead forecasts. As shown in Equation (1), the final forecast is a composition of level forecasts $\boldsymbol{E}_{t:t+H}$, growth representations $\boldsymbol{B}_{t:t+H}^{(n)}$ and seasonal representations $\boldsymbol{S}_{t:t+H}^{(n)}$ in the forecast horizon. It comprises $N$ Growth + Seasonal (G+S) Stacks, and a Level Stack. The G+S Stack consists of the Growth Damping (GD) and FA blocks, which leverage $\boldsymbol{B}_t^{(n)}, \boldsymbol{S}_{t-L:t}^{(n)}$ to predict $\boldsymbol{B}_{t:t+H}^{(n)}, \boldsymbol{S}_{t:t+H}^{(n)}$, respectively.

$$\text{Growth:} \quad \boldsymbol{B}_{t:t+H}^{(n)} = \text{GD}(\boldsymbol{B}_t^{(n)}) \qquad\qquad \text{Seasonal:} \quad \boldsymbol{S}_{t:t+H}^{(n)} = \text{FA}_{t:t+H}(\boldsymbol{S}_{t-L:t}^{(n)})$$

To obtain the level in the forecast horizon, the Level Stack repeats the level in the last time step $t$ along the forecast horizon. It can be defined as $\boldsymbol{E}_{t:t+H} = \text{Repeat}_H(\boldsymbol{E}_t^{(N)}) = [\boldsymbol{E}_t^{(N)}, \dots, \boldsymbol{E}_t^{(N)}]$, with $\text{Repeat}_H(\cdot) : \mathbb{R}^{1 \times m} \to \mathbb{R}^{H \times m}$.

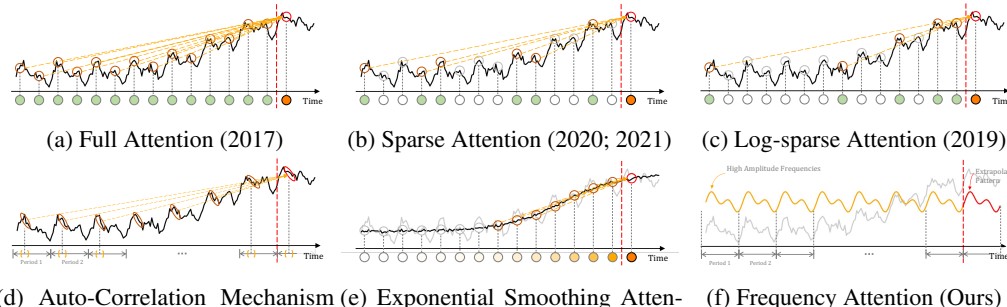

Figure 3: Comparison between different attention mechanisms. (a) Full, (b) Sparse, and (c) Log-sparse Attentions are adaptive mechanisms, where the green circles represent the attention weights adaptively calculated by a point-wise dot-product query, and depends on various factors including the time-series value, additional covariates (e.g. positional encodings, time features, etc.). (d) Auto-Correlation mechanism considers sliding dot-product queries to construct attention weights for each rolled input series. We introduce (e) Exponential Smoothing Attention (ESA) and (f) Frequency Attention (FA). ESA directly computes attention weights based on the relative time lag, without considering the input content, while FA attends to patterns which dominate with large magnitudes in the frequency domain.

**Growth Damping** To obtain the growth representation in the forecast horizon, we follow the idea of trend damping in Equation (4) to make robust multi-step forecast. Thus, the trend representations can be formulated as:

$$\mathrm{GD}(\boldsymbol{B}_t^{(n)})_j = \sum_{i=1}^{j} \gamma^i \boldsymbol{B}_t^{(n)},$$

$$\mathrm{GD}(\boldsymbol{B}_{t-L:t}^{(n)}) = [\mathrm{GD}(\boldsymbol{B}_t^{(n)})_t, \dots, \mathrm{GD}(\boldsymbol{B}_t^{(n)})_{t+H-1}],$$

where $0 < \gamma < 1$ is the damping parameter which is learnable, and in practice, we apply a multi-head version of trend damping by making use of $n_h$ damping parameters. Similar to the implementation for level forecast in the Level Stack, we only use the last trend representation in the lookback window $\boldsymbol{B}_t^{(n)}$ to forecast the trend representation in the forecast horizon.

## 2.2 Exponential Smoothing Attention and Frequency Attention Mechanism

Considering the ineffectiveness of existing attention mechanisms in handling time-series data, we develop the Exponential Smoothing Attention (ESA) and Frequency Attention (FA) mechanisms to extract latent growth and seasonal representations. ESA is a non-adaptive, learnable attention scheme with an inductive bias to attend more strongly to recent observations by following an exponential decay, while FA is a non-learnable attention scheme, that leverages Fourier transformation to select dominating seasonal patterns. A comparison between existing work and our proposed ESA and FA is illustrated in Figure 3.

### 2.2.1 Exponential Smoothing Attention

Vanilla self-attention can be regarded as a weighted combination of an input sequence, where the weights are normalized alignment scores measuring the similarity between input contents (Tsai et al., 2019). Inspired by the exponential smoothing in Equation (3), we aim to assign a higher weight to recent observations. It can be regarded as a novel form of attention whose weights are computed by the relative time lag, rather than input content. Thus, the ESA mechanism can be defined as $\mathcal{A}_{\mathrm{ES}} : \mathbb{R}^{L \times d} \to \mathbb{R}^{L \times d}$, where $\mathcal{A}_{\mathrm{ES}}(\boldsymbol{V})_t \in \mathbb{R}^d$ denotes the $t$-th row of the output matrix, representing the token corresponding to the $t$-th time step. Its exponential smoothing formula can be further written as:

$$\mathcal{A}_{\mathrm{ES}}(\boldsymbol{V})_t = \alpha \boldsymbol{V}_t + (1 - \alpha)\mathcal{A}_{\mathrm{ES}}(\boldsymbol{V})_{t-1} = \sum_{j=0}^{t-1} \alpha(1-\alpha)^j \boldsymbol{V}_{t-j} + (1-\alpha)^t \boldsymbol{v}_0,$$

where $0 < \alpha < 1$ and $\boldsymbol{v}_0$ are learnable parameters known as the smoothing parameter and initial state respectively.

**Efficient $\mathcal{A}_{\mathrm{ES}}$ algorithm** The straightforward implementation of the ESA mechanism by constructing the attention matrix, $\boldsymbol{A}_{\mathrm{ES}}$ and performing a matrix multiplication with the input sequence (detailed algorithm in Appendix B.4) results in an $\mathcal{O}(L^2)$ computational complexity.

$$\mathcal{A}_{\mathrm{ES}}(\boldsymbol{V}) = \begin{bmatrix} \mathcal{A}_{\mathrm{ES}}(\boldsymbol{V})_1 \\ \vdots \\ \mathcal{A}_{\mathrm{ES}}(\boldsymbol{V})_L \end{bmatrix} = \boldsymbol{A}_{\mathrm{ES}} \cdot \begin{bmatrix} \boldsymbol{v}_0^T \\ \boldsymbol{V} \end{bmatrix},$$

Yet, we are able to achieve an efficient algorithm by exploiting the unique structure of the exponential smoothing attention matrix, $\boldsymbol{A}_{\mathrm{ES}}$, which is illustrated in Appendix B.1. Each row of the attention matrix can be regarded as iteratively right shifting with padding (ignoring the first column). Thus, a matrix-vector multiplication can be computed with a cross-correlation operation, which in turn has an efficient fast Fourier transform implementation (Mathieu et al., 2014). The full algorithm is described in Algorithm 1, Appendix B.2, achieving an $\mathcal{O}(L \log L)$ complexity.

**Multi-Head Exponential Smoothing Attention (MH-ESA)** We use $\mathcal{A}_{\mathrm{ES}}$ as a basic building block, and develop the Multi-Head Exponential Smoothing Attention to extract latent growth representations. Formally, we obtain the growth representations by taking the successive difference of the residuals.

$$\tilde{\boldsymbol{Z}}_{t-L:t}^{(n)} = \mathrm{Linear}(\boldsymbol{Z}_{t-L:t}^{(n-1)}),$$
$$\boldsymbol{B}_{t-L:t}^{(n)} = \mathrm{MH}\text{-}\mathcal{A}_{\mathrm{ES}}(\tilde{\boldsymbol{Z}}_{t-L:t}^{(n)} - [\tilde{\boldsymbol{Z}}_{t-L:t-1}^{(n)}, \boldsymbol{v}_0^{(n)}]),$$
$$\boldsymbol{B}_{t-L:t}^{(n)} := \mathrm{Linear}(\boldsymbol{B}_{t-L:t}^{(n)}),$$

where MH-$\mathcal{A}_{\mathrm{ES}}$ is a multi-head version of $\mathcal{A}_{\mathrm{ES}}$ and $\boldsymbol{v}_0^{(n)}$ is the initial state from the ESA mechanism.

### 2.2.2 FREQUENCY ATTENTION

The goal of identifying and extracting seasonal patterns from the lookback window is twofold. Firstly, it can be used to perform de-seasonalization on the input signals such that downstream components are able to focus on modeling the level and growth information. Secondly, we are able to extrapolate the seasonal patterns to build representations for the forecast horizon. The main challenge is to automatically identify seasonal patterns. Fortunately, the use of power spectral density estimation for periodicity detection has been well studied (Vlachos et al., 2005). Inspired by these methods, we leverage the discrete Fourier transform (DFT, details in Appendix C) to develop the FA mechanism to extract dominant seasonal patterns.

Specifically, FA first decomposes input signals into their Fourier bases via a DFT along the temporal dimension, $\mathcal{F}(\boldsymbol{Z}_{t-L:t}^{(n-1)}) \in \mathbb{C}^{F \times d}$ where $F = \lfloor L/2 \rfloor + 1$, and selects bases with the $K$ largest amplitudes. An inverse DFT is then applied to obtain the seasonality pattern in time domain. Formally, this is given by the following equations:

$$\boldsymbol{\Phi}_{k,i} = \phi\Big(\mathcal{F}(\boldsymbol{Z}_{t-L:t}^{(n-1)})_{k,i}\Big), \quad \boldsymbol{A}_{k,i} = \Big|\mathcal{F}(\boldsymbol{Z}_{t-L:t}^{(n-1)})_{k,i}\Big|,$$
$$\kappa_i^{(1)}, \ldots, \kappa_i^{(K)} = \arg\underset{k \in \{2,\ldots,F\}}{\mathrm{Top\text{-}K}} \Big\{\boldsymbol{A}_{k,i}\Big\},$$
$$\boldsymbol{S}_{j,i}^{(n)} = \sum_{k=1}^{K} \boldsymbol{A}_{\kappa_i^{(k)},i}\Big[\cos(2\pi f_{\kappa_i^{(k)}} j + \boldsymbol{\Phi}_{\kappa_i^{(k)},i}) + \cos(2\pi \bar{f}_{\kappa_i^{(k)}} j + \bar{\boldsymbol{\Phi}}_{\kappa_i^{(k)},i})\Big], \tag{2}$$

where $\boldsymbol{\Phi}_{k,i}, \boldsymbol{A}_{k,i}$ are the phase/amplitude of the $k$-th frequency for the $i$-th dimension, $\arg \mathrm{Top\text{-}K}$ returns the arguments of the top $K$ amplitudes, $K$ is a hyperparameter, $f_k$ is the Fourier frequency of the corresponding index, and $\bar{f}_k, \bar{\boldsymbol{\Phi}}_{k,i}$ are the Fourier frequency/amplitude of the corresponding conjugates.

Finally, the latent seasonal representation of the $i$-th dimension for the lookback window is formulated as $\boldsymbol{S}_{t-L:t,i}^{(n)} = [\boldsymbol{S}_{t-L,i}^{(n)}, \ldots, \boldsymbol{S}_{t-1,i}^{(n)}]$. For the forecast horizon, the FA module extrapolates beyond the lookback window via, $\boldsymbol{S}_{t:t+H,i}^{(n)} = [\boldsymbol{S}_{t,i}^{(n)}, \ldots, \boldsymbol{S}_{t+H-1,i}^{(n)}]$. Since $K$ is a hyperparameter typically chosen for small values, the complexity for the FA mechanism is similarly $\mathcal{O}(L \log L)$.

Table 1: Multivariate forecasting results over various forecast horizons. Best results are **bolded**, and second best results are underlined.

| Methods | | ETSformer | | FEDformer | | Autoformer | | Informer | | LogTrans | | Reformer | | LSTnet | | ES-RNN | |
|---|---|---|---|---|---|---|---|---|---|---|---|---|---|---|---|---|---|
| Metrics | | MSE | MAE | MSE | MAE | MSE | MAE | MSE | MAE | MSE | MAE | MSE | MAE | MSE | MAE | MSE | MAE |
| ETTm2 | 96 | **0.189** | **0.280** | 0.203 | 0.287 | 0.255 | 0.339 | 0.365 | 0.453 | 0.768 | 0.642 | 0.658 | 0.619 | 3.142 | 1.365 | 0.204 | 0.323 |
| | 192 | **0.253** | **0.319** | 0.269 | 0.328 | 0.281 | 0.340 | 0.533 | 0.563 | 0.989 | 0.757 | 1.078 | 0.827 | 3.154 | 1.369 | 0.351 | 0.405 |
| | 336 | **0.314** | **0.357** | 0.325 | 0.366 | 0.339 | 0.372 | 1.363 | 0.887 | 1.334 | 0.872 | 1.549 | 0.972 | 3.160 | 1.369 | 0.476 | 0.485 |
| | 720 | **0.414** | **0.413** | 0.421 | 0.415 | 0.422 | 0.419 | 3.379 | 1.388 | 3.048 | 1.328 | 2.631 | 1.242 | 3.171 | 1.368 | 0.623 | 0.561 |
| ECL | 96 | 0.187 | 0.304 | **0.183** | **0.297** | 0.201 | 0.317 | 0.274 | 0.368 | 0.258 | 0.357 | 0.312 | 0.402 | 0.680 | 0.645 | 0.922 | 0.666 |
| | 192 | 0.199 | 0.315 | **0.195** | **0.308** | 0.222 | 0.334 | 0.296 | 0.386 | 0.266 | 0.368 | 0.348 | 0.433 | 0.725 | 0.676 | 0.499 | 0.479 |
| | 336 | 0.212 | 0.329 | **0.212** | **0.313** | 0.231 | 0.338 | 0.300 | 0.394 | 0.280 | 0.380 | 0.350 | 0.433 | 0.828 | 0.727 | 0.760 | 0.570 |
| | 720 | 0.233 | 0.345 | **0.231** | **0.343** | 0.254 | 0.361 | 0.373 | 0.439 | 0.283 | 0.376 | 0.340 | 0.420 | 0.957 | 0.811 | - | - |
| Exchange | 96 | **0.085** | **0.204** | 0.139 | 0.276 | 0.197 | 0.323 | 0.847 | 0.752 | 0.968 | 0.812 | 1.065 | 0.829 | 1.551 | 1.058 | 0.096 | 0.221 |
| | 192 | **0.182** | **0.303** | 0.256 | 0.369 | 0.300 | 0.369 | 1.204 | 0.895 | 1.040 | 0.851 | 1.188 | 0.906 | 1.477 | 1.028 | 0.214 | 0.360 |
| | 336 | **0.348** | **0.428** | 0.426 | 0.464 | 0.509 | 0.524 | 1.672 | 1.036 | 1.659 | 1.081 | 1.357 | 0.976 | 1.507 | 1.031 | 0.469 | 0.537 |
| | 720 | **1.025** | **0.774** | 1.090 | 0.800 | 1.447 | 0.941 | 2.478 | 1.310 | 1.941 | 1.127 | 1.510 | 1.016 | 2.285 | 1.243 | 1.997 | 1.143 |
| Traffic | 96 | 0.607 | 0.392 | **0.562** | **0.349** | 0.613 | 0.388 | 0.719 | 0.391 | 0.684 | 0.384 | 0.732 | 0.423 | 1.107 | 0.685 | 1.315 | 0.546 |
| | 192 | 0.621 | 0.399 | **0.562** | **0.346** | 0.616 | 0.382 | 0.696 | 0.379 | 0.685 | 0.390 | 0.733 | 0.420 | 1.157 | 0.706 | 0.727 | 0.373 |
| | 336 | 0.622 | 0.396 | **0.570** | **0.323** | 0.622 | 0.337 | 0.777 | 0.420 | 0.733 | 0.408 | 0.742 | 0.420 | 1.216 | 0.730 | - | - |
| | 720 | 0.632 | 0.396 | **0.596** | **0.368** | 0.660 | 0.408 | 0.864 | 0.472 | 0.717 | 0.396 | 0.755 | 0.423 | 1.481 | 0.805 | - | - |
| Weather | 96 | **0.197** | **0.281** | 0.217 | 0.296 | 0.266 | 0.336 | 0.300 | 0.384 | 0.458 | 0.490 | 0.689 | 0.596 | 0.594 | 0.587 | 0.585 | 0.507 |
| | 192 | **0.237** | **0.312** | 0.276 | 0.336 | 0.307 | 0.367 | 0.598 | 0.544 | 0.658 | 0.589 | 0.752 | 0.638 | 0.560 | 0.565 | 0.381 | 0.397 |
| | 336 | **0.298** | **0.353** | 0.339 | 0.380 | 0.359 | 0.359 | 0.578 | 0.523 | 0.797 | 0.652 | 0.639 | 0.596 | 0.597 | 0.587 | 0.628 | 0.533 |
| | 720 | **0.352** | **0.388** | 0.403 | 0.428 | 0.419 | 0.419 | 1.059 | 0.741 | 0.869 | 0.675 | 1.130 | 0.792 | 0.618 | 0.599 | 0.711 | 0.545 |
| ILI | 24 | 2.527 | 1.020 | **2.203** | **0.963** | 3.483 | 1.287 | 5.764 | 1.677 | 4.480 | 1.444 | 4.400 | 1.382 | 6.026 | 1.770 | 5.393 | 1.561 |
| | 36 | 2.615 | 1.007 | **2.272** | **0.976** | 3.103 | 1.148 | 4.755 | 1.467 | 4.799 | 1.467 | 4.783 | 1.448 | 5.340 | 1.668 | 6.478 | 1.751 |
| | 48 | 2.359 | **0.972** | **2.209** | 0.981 | 2.669 | 1.085 | 4.763 | 1.469 | 4.800 | 1.468 | 4.832 | 1.465 | 6.080 | 1.787 | 7.160 | 1.963 |
| | 60 | **2.487** | **1.016** | 2.545 | 1.061 | 2.770 | 1.125 | 5.264 | 1.564 | 5.278 | 1.560 | 4.882 | 1.483 | 5.548 | 1.720 | 5.801 | 1.711 |

## 3 EXPERIMENTS

This section presents extensive empirical evaluations on the LSTF task over 6 real world multivariate datasets, ETT, ECL, Exchange, Traffic, Weather, and ILI, coming from a variety of application areas (details in Appendix E). Performance is evaluated via the mean squared error (MSE) and mean absolute error (MAE) metrics. For the main benchmark, datasets are split into train, validation, and test sets chronologically, following a 60/20/20 split for the ETT datasets and 70/10/20 split for other datasets. The multivariate benchmark makes use of all dimensions, while univariate benchmark selects the last dimension of the datasets as the target variable, following previous work (Zhou et al., 2021; Wu et al., 2021). Data is pre-processed by performing standardization based on train set statistics. Further details on implementation and hyperparameters can be found in Appendix D. This is followed by an ablation study of the various contributing components, and interpretability experiments of our proposed model, and finally an analysis on computational efficiency.

### 3.1 RESULTS

For the multivariate benchmark, baselines include recently proposed time-series/efficient Transformers – FEDformer, Autoformer, Informer, LogTrans (Li et al., 2019), and Reformer (Kitaev et al., 2020), and RNN variants – LSTnet (Lai et al., 2018), and ES-RNN (Smyl, 2020). Univariate baselines further include N-BEATS (Oreshkin et al., 2019), DeepAR (Salinas et al., 2020), ARIMA, Prophet (Taylor & Letham, 2018), and AutoETS (Bhatnagar et al., 2021). We obtain baseline results from the following papers: (Wu et al., 2021; Zhou et al., 2021), and further run AutoETS from the Merlion library (Bhatnagar et al., 2021). Table 1 summarize the results of ETSformer against top performing baselines on a selection of datasets, for the multivariate setting, and Table 6 in Appendix G for space. Results for ETSformer are averaged over three runs (standard deviation in Appendix H). Overall, ETSformer achieves competitive performance, achieving best performance on 14 out of 24 datasets/settings on MSE for the multivariate case, and within top 2 performance across all 24 datasets/settings.

### 3.2 ABLATION STUDY

We study the contribution of each major component which the final forecast is composed of level, growth, and seasonality. Table 2 first presents the performance of the full model, and subsequently, the performance of the resulting model by removing each component. We observe that the composition of level, growth, and season provides the most accurate forecasts across a variety of application areas, and removing any one component results in a deterioration. In particular, estimation of the level of the time-series is critical. We also analyse the design of the MH-ESA in Section 3.2, replacing it with a vanilla multi-head attention and an FC layer performing token mixing – we observe that our trend attention formulation indeed is more effective.

Table 2: Ablation study on the various components (Level, Growth, Season) of ETSformer, averaged over multiple horizons {24, 96, 192, 336, 720} for ETTm2, ECL, and Traffic, {24, 36, 48, 60} for ILI.

| Datasets | | ETTm2 | ECL | Traffic | ILI |
|---|---|---|---|---|---|
| ETSformer | MSE | **0.256** | **0.199** | **0.611** | **2.570** |
| | MAE | **0.318** | **0.316** | **0.391** | **1.029** |
| w/o Level | MSE | 2.426 | 0.306 | 0.683 | 4.994 |
| | MAE | 1.146 | 0.396 | 0.412 | 1.628 |
| w/o Season | MSE | 0.302 | 0.819 | 1.393 | 4.110 |
| | MAE | 0.348 | 0.745 | 0.792 | 1.437 |
| w/o Growth | MSE | 0.261 | 0.202 | 0.619 | 2.642 |
| | MAE | 0.321 | 0.317 | 0.397 | 1.101 |

Table 3: Ablation study on the effectiveness of the MH-ESA design.

| Datasets | | ETTm2 | ECL | Traffic | ILI |
|---|---|---|---|---|---|
| ETSformer | MSE | **0.256** | **0.199** | **0.611** | **2.570** |
| | MAE | **0.318** | **0.316** | **0.391** | **1.029** |
| MH-ESA → MHA | MSE | 0.548 | 0.239 | 0.632 | 3.408 |
| | MAE | 0.570 | 0.262 | 0.591 | 2.485 |
| MH-ESA → FC | MSE | 0.342 | 0.235 | 0.626 | 2.779 |
| | MAE | 0.394 | 0.348 | 0.395 | 1.062 |

Table 4: MSE of decomposed forecasts over the synthetic dataset's test set (1000 samples).

| | Combined | Trend | Season |
|---|---|---|---|
| ETSformer | 0.009 | 0.003 | 0.005 |
| Autoformer | 0.037 | 0.046 | 0.008 |
| FEDformer | 0.012 | 0.201 | 0.198 |

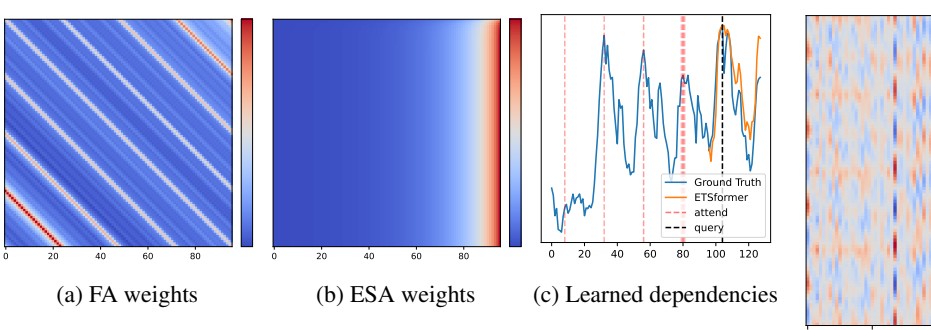

(a) FA weights     (b) ESA weights     (c) Learned dependencies

Figure 4: ETSformer attention weights visualization and learned seasonal dependencies on the ECL dataset. For weights visualizations, each row represents the attention weights a time step in the forecast horizon places on each time step in the lookback window. FA learns a clear periodicity, which is highlighted in the learned dependencies, where the top 6 time steps being attended to by the query time step are highlighted in red. ESA displays exponentially decaying weights representing growth.

Figure 5: Autoformer Auto-Correlation mechanism weights on ECL dataset.

### 3.3 INTERPRETABILITY

ETSformer generates interpretable forecasts which can be decomposed into disentangled level, growth, and seasonal components. We showcased this ability compared to baselines in Figure 1 on synthetic data containing (nonlinear) trend and seasonality patterns (details in Appendix F) , since we are not able to obtain ground truth decomposition from real-world data. Forecast decompositions (without component ground truth) can be found in Appendix J. Furthermore, we report quantitative results over the test set in Table 4. ETSformer successfully forecasts interpretable level, trend (level + growth), and seasonal components, as observed in the trend and seasonality components closely tracking the ground truth patterns. Despite obtaining a good combined forecast, competing decomposition based approaches, struggles to disambiguate between trend and seasonality.

Furthermore, ETSformer produces human interpretable attention weights for both the FA and ESA mechanisms, visualized in Figure 4. The FA weights visualized exhibit clear periodicity which can be used to identify the dominating seasonal patterns, while ESA weights exhibit exponentially decaying property as per the inductive biases. This is contrasted to Autoformer's Auto-Correlation visualization in Figure 5 which does not follow periodicity properties despite being specialized to handle seasonality.

### 3.4 COMPUTATIONAL EFFICIENCY

Figure 6 charts ETSformer's empirical efficiency with that of competing Transformer-based approaches. ETSformer maintains competitive efficiency with competing quasilinear and linear complexity Transformers. This is especially so when forecast horizon increasese, due to ETSformer's unique decoder architecture which relies on its Trend Damping and Frequency Attention modules rather than relying on a cross attention mechanism. Of note, while FEDformer claims linear complexity, our empirical results show that it incurs significant overhead especially in terms of runtime efficiency. This slowdown arises from their (official) implementation still relying on the straightfor-

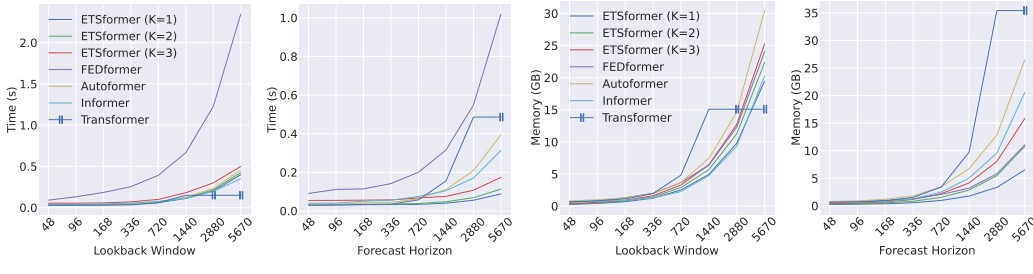

(a) Runtime Efficiency Analysis           (b) Memory Efficiency Analysis

Figure 6: Computational Efficiency Analysis. Values reported are based on the training phase of ETTm2 multivariate setting. Horizon is fixed to 48 for lookback window plots, and lookback is fixed to 48 for forecast horizon plots. For runtime efficiency, values refer to the time for one iteration. The "‖" marker indicates an out-of-memory error for those settings.

ward FFT operation, incurring $\mathcal{O}(L \log L)$ complexity, as well as their Frequency Enhanced Modules requiring a large number of trainable parameters.

## 4   RELATED WORK

**Deep Forecasting**   LogTrans (Li et al., 2019) and AST (Wu et al., 2020) first introduced Transformer based methods to reduce computational complexity of attention. The LSTF benchmark was first introduced by Informer (Zhou et al., 2021), extending the Transformer architecture by proposing the ProbSparse attention and distillation operation to achieve $\mathcal{O}(L \log L)$ complexity. Similar to our work that incorporates prior knowledge of time-series structure, Autoformer (Wu et al., 2021) introduces the Auto-Correlation attention mechanism which focuses on sub-series based similarity and is able to extract periodic patterns. FEDformer (Zhou et al., 2022) extends this line of work by incorporating Frequency Enhanced structures. N-HiTS (Challu et al., 2022) introduced hierarchical interpolation and multi-rate data sampling by building on top of N-BEATS (Oreshkin et al., 2019) for the LSTF task. ES-RNN (Smyl, 2020) has explored combining ETS methods with neural networks. However, they treat ETS as a pre and post processing step, rather than baking it into the model architecture. Furthermore, their method requires prior knowledge on seasonality patterns, and they were not proposed for LSTF, leading to high computation costs over long horizons.

**Attention Mechanisms**   The self-attention mechanism in Transformer models has recently received much attention, its necessity has been greatly investigated in attempts to introduce more flexibility and reduce computational cost. Synthesizer (Tay et al., 2021) empirically studies the importance of dot-product interactions, and show that a randomly initialized, learnable attention mechanisms with or without token-token dependencies can achieve competitive performance with vanilla self-attention on various NLP tasks. You et al. (2020) utilizes an unparameterized Gaussian distribution to replace the original attention scores, concluding that the attention distribution should focus on a certain local window and can achieve comparable performance. Raganato et al. (2020) replaces attention with fixed, non-learnable positional patterns, obtaining competitive performance on NMT tasks. Lee-Thorp et al. (2021) replaces self-attention with a non-learnable Fourier Transform and verifies it to be an effective mixing mechanism.

## 5   DISCUSSION

Inspired by the classical exponential smoothing methods and emerging Transformer approaches for time-series forecasting, we propose ETSformer, a novel level-growth-season decomposition Transfomer. ETSformer leverages the novel Exponential Smoothing Attention and Frequency Attention mechanisms which are more effective at modeling time-series than vanilla self-attention, and at the same time achieves $\mathcal{O}(L \log L)$ complexity, where $L$ is the length of lookback window. We performed extensive empirical evaluation, showing that ETSformer has extremely competitive accuracy and efficiency, while being highly interpretable.

**Limitations & Future Work**   ETSformer currently only produces point forecasts. Probabilistic forecasting would be a valuable extension of our current work due to it's importance in practical applications. Other future directions which ETSformer does not currently consider but would be useful are additional covariates such as holiday indicators and other dummy variables to consider holiday effects which cannot be captured by the FA mechanism.

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

## A CLASSICAL EXPONENTIAL SMOOTHING

We instantiate exponential smoothing methods (Hyndman et al., 2008) in the univariate forecasting setting. They assume that time-series can be decomposed into seasonal and trend components, and trend can be further decomposed into level and growth components. Specifically, a commonly used model is the additive Holt-Winters' method (Holt, 2004; Winters, 1960), which can be formulated as:

$$
\begin{aligned}
\text{Level} &: e_t = \alpha(x_t - s_{t-p}) + (1 - \alpha)(e_{t-1} + b_{t-1}) \\
\text{Growth} &: b_t = \beta(e_t - e_{t-1}) + (1 - \beta)b_{t-1} \\
\text{Seasonal} &: s_t = \gamma(x_t - e_t) + (1 - \gamma)s_{t-p} \\
\text{Forecasting} &: \hat{x}_{t+h|t} = e_t + hb_t + s_{t+h-p}
\end{aligned}
\tag{3}
$$

where $p$ is the period of seasonality, and $\hat{x}_{t+h|t}$ is the $h$-steps ahead forecast. The above equations state that the $h$-steps ahead forecast is composed of the last estimated level $e_t$, incrementing it by $h$ times the last growth factor, $b_t$, and adding the last available seasonal factor $s_{t+h-p}$. Specifically, the level smoothing equation is formulated as a weighted average of the seasonally adjusted observation $(x_t - s_{t-p})$ and the non-seasonal forecast, obtained by summing the previous level and growth $(e_{t-1} + b_{t-1})$. The growth smoothing equation is implemented by a weighted average between the successive difference of the (de-seasonalized) level, $(e_t - e_{t-1})$, and the previous growth, $b_{t-1}$. Finally, the seasonal smoothing equation is a weighted average between the difference of observation and (de-seasonalized) level, $(x_t - e_t)$, and the previous seasonal index $s_{t-p}$. The weighted average of these three equations are controlled by the smoothing parameters $\alpha$, $\beta$ and $\gamma$, respectively.

A widely used modification of the additive Holt-Winters' method is to allow the damping of trends, which has been proved to produce robust multi-step forecasts (Svetunkov, 2016; McKenzie & Gardner Jr, 2010). The forecast with damping trend can be rewritten as:

$$
\hat{x}_{t+h|t} = e_t + (\phi + \phi^2 + \cdots + \phi^h)b_t + s_{t+h-p},
\tag{4}
$$

where the growth is damped by a factor of $\phi$. If $\phi = 1$, it degenerates to the vanilla forecast. For $0 < \phi < 1$, as $h \to \infty$ this growth component approaches an asymptote given by $\phi b_t/(1 - \phi)$.

## B EXPONENTIAL SMOOTHING ATTENTION

### B.1 EXPONENTIAL SMOOTHING ATTENTION MATRIX

$$
\boldsymbol{A}_{\text{ES}} =
\begin{bmatrix}
(1-\alpha)^1 & \alpha & 0 & 0 & \dots & 0 \\
(1-\alpha)^2 & \alpha(1-\alpha) & \alpha & 0 & \dots & 0 \\
(1-\alpha)^3 & \alpha(1-\alpha)^2 & \alpha(1-\alpha) & \alpha & \dots & 0 \\
\vdots & \vdots & \vdots & \vdots & \ddots & \vdots \\
(1-\alpha)^L & \alpha(1-\alpha)^{L-1} & \dots & \alpha(1-\alpha)^j & \dots & \alpha
\end{bmatrix}
$$

### B.2 EFFICIENT EXPONENTIAL SMOOTHING ATTENTION ALGORITHM

---

**Algorithm 1** PyTorch-style pseudocode of efficient $\mathcal{A}_{\text{ES}}$

---

`conv1d_fft`: efficient convolution operation implemented with fast Fourier transform (Appendix B, Algorithm 3), `outer`: outer product

```
# V: value matrix, shape: L x d
# v0: initial state, shape: d
# alpha: smoothing parameter, shape: 1

# obtain exponentially decaying weights
# and compute weighted combination
powers = arange(L) # L
weight = alpha * (1 - alpha) ** flip(powers) # L
output = conv1d_fft(V, weight, dim=0) # L x d

# compute contribution from initial state
init_weight = (1 - alpha) ** (powers + 1) # L
init_output = outer(init_weight, v0) # L x d
return init_output + output
```

---

### B.3 LEVEL SMOOTHING VIA EXPONENTIAL SMOOTHING ATTENTION

$$
\begin{aligned}
\boldsymbol{E}_t^{(n)} &= \boldsymbol{\alpha} * (\boldsymbol{E}_t^{(n-1)} - \boldsymbol{S}_t^{(n)}) + (1 - \boldsymbol{\alpha}) * (\boldsymbol{E}_{t-1}^{(n)} + \boldsymbol{B}_{t-1}^{(n)}) \\
&= \boldsymbol{\alpha} * (\boldsymbol{E}_t^{(n-1)} - \boldsymbol{S}_t^{(n)}) + (1 - \boldsymbol{\alpha}) * \boldsymbol{B}_{t-1}^{(n)} \\
&\quad + (1 - \boldsymbol{\alpha}) * [\boldsymbol{\alpha} * (\boldsymbol{E}_{t-1}^{(n-1)} - \boldsymbol{S}_{t-1}^{(n)}) + (1 - \boldsymbol{\alpha}) * (\boldsymbol{E}_{t-2}^{(n)} + \boldsymbol{B}_{t-2}^{(n)})] \\
&= \boldsymbol{\alpha} * (\boldsymbol{E}_t^{(n-1)} - \boldsymbol{S}_t^{(n)}) + \boldsymbol{\alpha} * (1 - \boldsymbol{\alpha}) * (\boldsymbol{E}_{t-1}^{(n-1)} - \boldsymbol{S}_{t-1}^{(n)}) \\
&\quad + (1 - \boldsymbol{\alpha}) * \boldsymbol{B}_{t-1}^{(n)} + (1 - \boldsymbol{\alpha})^2 * \boldsymbol{B}_{t-2}^{(n)} \\
&\quad + (1 - \boldsymbol{\alpha})^2 [\boldsymbol{\alpha} * (\boldsymbol{E}_{t-2}^{(n-1)} - \boldsymbol{S}_{t-2}^{(n)}) + (1 - \boldsymbol{\alpha}) * (\boldsymbol{E}_{t-3}^{(n)} + \boldsymbol{B}_{t-3}^{(n)})] \\
&\ \ \vdots \\
&= (1 - \boldsymbol{\alpha})^t (\boldsymbol{E}_0^{(n)} - \boldsymbol{S}_0^{(n)}) + \sum_{j=0}^{t-1} \boldsymbol{\alpha} * (1 - \boldsymbol{\alpha})^j * (\boldsymbol{E}_{t-j}^{(n-1)} - \boldsymbol{S}_{t-j}^{(n)}) + \sum_{k=1}^{t} (1 - \boldsymbol{\alpha})^k * \boldsymbol{B}_{t-k}^{(n)} \\
&= \mathcal{A}_{\mathrm{ES}}(\boldsymbol{E}_{t-L:t}^{(n-1)} - \boldsymbol{S}_{t-L:t}^{(n)}) + \sum_{k=1}^{t} (1 - \boldsymbol{\alpha})^k * \boldsymbol{B}_{t-k}^{(n)}
\end{aligned}
$$

Based on the above expansion of the level equation, we observe that $\boldsymbol{E}_n^{(t)}$ can be computed by a sum of two terms, the first of which is given by an $\mathcal{A}_{\mathrm{ES}}$ term, and we finally, we note that the second term can also be calculated using the conv1d_fft algorithm, resulting in a fast implementation of level smoothing.

## B.4 Further Details on ESA Implementation

**Algorithm 2** PyTorch-style pseudocode of naive $\mathcal{A}_{\text{ES}}$

```
mm: matrix multiplication, outer: outer product
repeat: einops style tensor operations,
gather: gathers values along an axis specified by dim

# V: value matrix, shape: L x d
# v0: initial state, shape: d
# alpha: smoothing parameter, shape: 1

L, d = V.shape

# obtain exponentially decaying weights
powers = arange(L) # L
weight = alpha * (1 - alpha).pow(flip(powers)) #
    L

# perform a strided roll operation
# rolls a matrix along the columns in a strided
    manner
# i.e. first row is shifted right by L-1
    positions,
# second row is shifted L-2, ..., last row is
    shifted by 0.
weight = repeat(weight, 'L -> T L', T=L) # L x L
indices = repeat(arange(L), 'L -> T L', T=L)
indices = (indices - (arange(L) + 1).unsqueeze(1)
    ) % L
weight = gather(weight, dim=-1, index=indices)

# triangle masking to achieve the exponential
    smoothing attention matrix
weight = triangle_causal_mask(weight)

output = mm(weight, V)

init_weight = (1 - alpha) ** (powers + 1)
init_output = outer(init_weight, v0)

return init_output + output
```

**Algorithm 3** PyTorch-style pseudocode of conv1d_fft

```
next_fast_len: find the next fast size of input data to fft,
for zero-padding, etc.
rfft: compute the one-dimensional discrete Fourier Trans-
form for real input
x.conj(): return the complex conjugate, element-wise
irfft: computes the inverse of rfft
roll: roll array elements along a given axis
index_select: returns a new tensor which index_es the
input tensor along dimension dim using the entries in index

# V: value matrix, shape: L x d
# weight: exponential smoothing attention vector,
      shape: L
# dim: dimension to perform convolution on

# obtain lengths of sequence to perform
    convolution on
N = V.size(dim)
M = weight.size(dim)

# Fourier transform on inputs
fast_len = next_fast_len(N + M - 1)
F_V = rfft(V, fast_len, dim=dim)
F_weight = rfft(weight, fast_len, dim=dim)

# multiplication and inverse
F_V_weight = F_V * F_weight.conj()
out = irfft(F_V_weight, fast_len, dim=dim)
out = out.roll(-1, dim=dim)

# select the correct indices
idx = range(fast_len - N, fast_len)
out = out.index_select(dim, idx)

return out
```

Algorithm 2 describes the naive implementation for ESA by first constructing the exponential smoothing attention matrix, $\boldsymbol{A}_{\text{ES}}$, and performing the full matrix-vector multiplication. Efficient $\mathcal{A}_{\text{ES}}$ relies on Algorithm 3, to achieve an $\mathcal{O}(L \log L)$ complexity, by speeding up the matrix-vector multiplication. Due to the structure lower triangular structure of $\boldsymbol{A}_{\text{ES}}$ (ignoring the first column), we note that performing a matrix-vector multiplication with it is equivalent to performing a convolution with the last row. Algorithm 3 describes the pseudocode for fast convolutions using fast Fourier transforms.

## C Discrete Fourier Transform

The DFT of a sequence with regular intervals, $\boldsymbol{x} = (x_0, x_1, \ldots, x_{N-1})$ is a sequence of complex numbers,

$$c_k = \sum_{n=0}^{N-1} x_n \cdot \exp(-i2\pi kn/N),$$

for $k = 0, 1, \ldots, N-1$, where $c_k$ are known as the Fourier coefficients of their respective Fourier frequencies. Due to the conjugate symmetry of DFT for real-valued signals, we simply consider the first $\lfloor N/2 \rfloor + 1$ Fourier coefficients and thus we denote the DFT as $\mathcal{F} : \mathbb{R}^N \to \mathbb{C}^{\lfloor N/2 \rfloor + 1}$. The DFT maps a signal to the frequency domain, where each Fourier coefficient can be uniquely represented

by the amplitude, $|c_k|$, and the phase, $\phi(c_k)$,

$$|c_k| = \sqrt{\Re\{c_k\}^2 + \Im\{c_k\}^2} \qquad\qquad \phi(c_k) = \tan^{-1}\left(\frac{\Im\{c_k\}}{\Re\{c_k\}}\right)$$

where $\Re\{c_k\}$ and $\Im\{c_k\}$ are the real and imaginary components of $c_k$ respectively. Finally, the inverse DFT maps the frequency domain representation back to the time domain,

$$x_n = \mathcal{F}^{-1}(\boldsymbol{c})_n = \frac{1}{N}\sum_{k=0}^{N-1} c_k \cdot \exp(i2\pi kn/N),$$

## D IMPLEMENTATION DETAILS

### D.1 HYPERPARAMETERS

For all experiments, we use the same hyperparameters for the encoder layers, decoder stacks, model dimensions, feedforward layer dimensions, number of heads in multi-head exponential smoothing attention, and kernel size for input embedding as listed in Table 5. We perform hyperparameter tuning via a grid search over the number of frequencies $K$, lookback window size, and learning rate, selecting the settings which perform the best on the validation set based on MSE (on results averaged over three runs). The search range is reported in Table 5, where the lookback window size search range was decided to be set as the values for the horizon sizes for the respective datasets.

Table 5: Hyperparameters used in ETSformer.

| Hyperparameter | Value |
|---|---|
| Encoder layers | 2 |
| Decoder stacks | 2 |
| Model dimension | 512 |
| Feedforward dimension | 2048 |
| Multi-head ESA heads | 8 |
| Input embedding kernel size | 3 |
| K | $K \in \{0, 1, 2, 3\}$ |
| Lookback window size | $L \in \{96, 192, 336, 720\}$ |
| Lookback window size (ILI) | $L \in \{24, 36, 48, 60\}$ |
| Learning rate | $lr \in \{1\mathrm{e}-3, 3\mathrm{e}-4, 1\mathrm{e}-4, 3\mathrm{e}-5, 1\mathrm{e}-5\}$ |

### D.2 OPTIMIZATION

We use the Adam optimizer (Kingma & Ba, 2015) with $\beta_1 = 0.9$, $\beta_2 = 0.999$, and $\epsilon = 1e-08$, and a batch size of 32. We schedule the learning rate with linear warmup over 3 epochs, and cosine annealing thereafter for a total of 15 training epochs for all datasets. The minimum learning rate is set to 1e-30. For smoothing and damping parameters, we set the learning rate to be 100 times larger and do not use learning rate scheduling. Training was done on an Nvidia A100 GPU.

### D.3 REGULARIZATION

We apply two forms of regularization during the training phase.

**Dropout**  We apply dropout (Srivastava et al., 2014) with a rate of $p = 0.2$ across the model. Dropout is applied on the outputs of the Input Embedding, Frequency Self-Attention and Multi-Head ES Attention blocks, in the Feedforward block (after activation and before normalization), on the attention weights, as well as damping weights.

**Noise Injection**  We utilize a composition of three noise distributions, applied in the following order - scale, shift, and jitter, activating with a probability of 0.5.

1. Scale – The time-series is scaled by a single random scalar value, obtained by sampling $\epsilon \sim \mathcal{N}(0, 0.2)$, and each time step is $\tilde{x}_t = \epsilon x_t$.

2. Shift – The time-series is shifted by a single random scalar value, obtained by sampling $\epsilon \sim \mathcal{N}(0, 0.2)$ and each time step is $\tilde{x}_t = x_t + \epsilon$.

3. Jitter – I.I.D. Gaussian noise is added to each time step, from a distribution $\epsilon_t \sim \mathcal{N}(0, 0.2)$, where each time step is now $\tilde{x}_t = x_t + \epsilon_t$.

## E    DATASETS

**ETT**[1] Electricity Transformer Temperature (Zhou et al., 2021) is a multivariate time-series dataset, comprising of load and oil temperature data recorded every 15 minutes from electricity transformers. ETT consists of two variants, ETTm and ETTh, whereby ETTh is the hourly-aggregated version of ETTm, the original 15 minute level dataset.

**ECL**[2] Electricity Consuming Load measures the electricity consumption of 321 households clients over two years, the original dataset was collected at the 15 minute level, but is pre-processed into an hourly level dataset.

**Exchange**[3] Exchange (Lai et al., 2018) tracks the daily exchange rates of eight countries (Australia, United Kingdom, Canada, Switzerland, China, Japan, New Zealand, and Singapore) from 1990 to 2016.

**Traffic**[4] Traffic is an hourly dataset from the California Department of Transportation describing road occupancy rates in San Francisco Bay area freeways.

**Weather**[5] Weather measures 21 meteorological indicators like air temperature, humidity, etc., every 10 minutes for the year of 2020.

**ILI**[6] Influenza-like Illness records the ratio of patients seen with ILI and the total number of patients on a weekly basis, obtained by the Centers for Disease Control and Prevention of the United States between 2002 and 2021.

## F    SYNTHETIC DATASET

The synthetic dataset is constructed by a combination of trend and seasonal component. Each instance in the dataset has a lookack window length of 720 and forecast horizon length of 192. The trend pattern follows a nonlinear, saturating pattern, $b(t) = \frac{1}{1 + \exp \beta_0 (t - \beta_1)}$, where $\beta_0 = -0.2, \beta_1 = 720$. The seasonal pattern follows a complex periodic pattern formed by a sum of sinusoids. Concretely, $s(t) = A_1 \cos(2\pi f_1 t) + A_2 \cos(2\pi f_2 t$, where $f_1 = 1/10, f_2 = 1/13$ are the frequencies, $A_1 = A_2 = 0.15$ are the amplitudes. During training phase, we use an additional noise component by adding i.i.d. gaussian noise with 0.05 standard deviation. Finally, the $i$-th instance of the dataset is $x_i = [x_i(1), x_i(2), \ldots, x_i(720 + 192)]$, where $x_i(t) = b(t) + s(t + i)$.

---

[1] https://github.com/zhouhaoyi/ETDataset
[2] lhttps://archive.ics.uci.edu/ml/datasets/ElectricityLoadDiagrams20112014
[3] https://github.com/laiguokun/multivariate-time-series-data
[4] https://pems.dot.ca.gov/
[5] https://www.bgc-jena.mpg.de/wetter/
[6] https://gis.cdc.gov/grasp/fluview/fluportaldashboard.html

## G    UNIVARIATE FORECASTING BENCHMARK

Table 6: Univariate forecasting results over various forecast horizons. Best results are **bolded**, and second best results are underlined.

| Methods | | ETSformer | | FEDformer | | Autoformer | | Informer | | N-BEATS | | DeepAR | | Prophet | | ARIMA | | AutoETS | |
|---|---|---|---|---|---|---|---|---|---|---|---|---|---|---|---|---|---|---|---|---|
| Metrics | | MSE | MAE | MSE | MAE | MSE | MAE | MSE | MAE | MSE | MAE | MSE | MAE | MSE | MAE | MSE | MAE | MSE | MAE |
| ETTm2 | 96 | 0.080 | 0.212 | **0.063** | **0.189** | 0.065 | **0.189** | 0.088 | 0.225 | 0.082 | 0.219 | 0.099 | 0.237 | 0.287 | 0.456 | 0.211 | 0.362 | 0.794 | 0.617 |
| | 192 | 0.150 | 0.302 | **0.102** | **0.245** | 0.118 | 0.256 | 0.132 | 0.283 | 0.120 | 0.268 | 0.154 | 0.310 | 0.312 | 0.483 | 0.261 | 0.406 | 1.078 | 0.740 |
| | 336 | 0.175 | 0.334 | **0.130** | **0.279** | 0.154 | 0.305 | 0.180 | 0.336 | 0.226 | 0.370 | 0.277 | 0.428 | 0.331 | 0.474 | 0.317 | 0.448 | 1.279 | 0.822 |
| | 720 | 0.224 | 0.379 | **0.178** | **0.325** | 0.182 | 0.335 | 0.300 | 0.435 | 0.188 | 0.338 | 0.332 | 0.468 | 0.534 | 0.593 | 0.366 | 0.487 | 1.541 | 0.924 |
| Exchange | 96 | **0.099** | **0.230** | 0.131 | 0.284 | 0.241 | 0.299 | 0.591 | 0.615 | 0.156 | 0.299 | 0.417 | 0.515 | 0.828 | 0.762 | 0.112 | 0.245 | 0.192 | 0.316 |
| | 192 | **0.223** | **0.353** | 0.277 | 0.420 | 0.273 | 0.665 | 1.183 | 0.912 | 0.669 | 0.665 | 0.813 | 0.735 | 0.909 | 0.974 | 0.304 | 0.404 | 0.355 | 0.442 |
| | 336 | **0.421** | **0.497** | 0.426 | 0.511 | 0.508 | 0.605 | 1.367 | 0.984 | 0.611 | 0.605 | 1.331 | 0.962 | 1.304 | 0.988 | 0.736 | 0.598 | 0.577 | 0.578 |
| | 720 | 1.114 | **0.807** | 1.162 | 0.832 | **0.991** | 0.860 | 1.872 | 1.072 | 1.111 | 0.860 | 1.890 | 1.181 | 3.238 | 1.566 | 1.871 | 0.935 | 1.242 | 0.865 |

## H    ETSFORMER STANDARD DEVIATION

Table 7: ETSformer main benchmark results with standard deviation. Experiments are performed over three runs.

(a) Multivariate benchmark.

| Metrics | | MSE (SD) | MAE (SD) |
|---|---|---|---|
| ETTm2 | 96 | 0.189 (0.002) | 0.280 (0.001) |
| | 192 | 0.253 (0.002) | 0.319 (0.001) |
| | 336 | 0.314 (0.001) | 0.357 (0.001) |
| | 720 | 0.414 (0.000) | 0.413 (0.001) |
| ECL | 96 | 0.187 (0.001) | 0.304 (0.001) |
| | 192 | 0.199 (0.001) | 0.315 (0.002) |
| | 336 | 0.212 (0.001) | 0.329 (0.002) |
| | 720 | 0.233 (0.006) | 0.345 (0.006) |
| Exchange | 96 | 0.085 (0.000) | 0.204 (0.001) |
| | 192 | 0.182 (0.003) | 0.303 (0.002) |
| | 336 | 0.348 (0.004) | 0.428 (0.003) |
| | 720 | 1.025 (0.031) | 0.774 (0.014) |
| Traffic | 96 | 0.607 (0.005) | 0.392 (0.005) |
| | 192 | 0.621 (0.015) | 0.399 (0.013) |
| | 336 | 0.622 (0.003) | 0.396 (0.003) |
| | 720 | 0.632 (0.004) | 0.396 (0.004) |
| Weather | 96 | 0.197 (0.007) | 0.281 (0.008) |
| | 192 | 0.237 (0.005) | 0.312 (0.004) |
| | 336 | 0.298 (0.003) | 0.353 (0.003) |
| | 720 | 0.352 (0.007) | 0.388 (0.002) |
| ILI | 24 | 2.527 (0.061) | 1.020 (0.021) |
| | 36 | 2.615 (0.103) | 1.007 (0.013) |
| | 48 | 2.359 (0.056) | 0.972 (0.011) |
| | 60 | 2.487 (0.006) | 1.016 (0.007) |

(b) Univariate benchmark.

| Metrics | | MSE (SD) | MAE (SD) |
|---|---|---|---|
| ETTm2 | 96 | 0.080 (0.001) | 0.212 (0.001) |
| | 192 | 0.150 (0.024) | 0.302 (0.026) |
| | 336 | 0.175 (0.012) | 0.334 (0.014) |
| | 720 | 0.224 (0.008) | 0.379 (0.006) |
| Exchange | 96 | 0.099 (0.003) | 0.230 (0.003) |
| | 192 | 0.223 (0.015) | 0.353 (0.009) |
| | 336 | 0.421 (0.002) | 0.497 (0.000) |
| | 720 | 1.114 (0.049) | 0.807 (0.016) |

# I  LAYER ANALYSIS

Table 8: Analysis on the number of layers and stacks of ETSformer for ETTm2 and ECL datasets. Obs Space refers to a variation of ETSformer which removes the embedding projection layer, and performs operations in observation space.

| Num Layers | | Obs. Space | | Layers=1 | | Layers=2 | | Layers=3 | | Layers=4 | | Layers=5 | |
|---|---|---|---|---|---|---|---|---|---|---|---|---|---|
| Metrics | | MSE | MAE | MSE | MAE | MSE | MAE | MSE | MAE | MSE | MAE | MSE | MAE |
| ETTm2 | 96 | 0.685 | 0.705 | 0.190 | 0.284 | 0.189 | 0.280 | **0.188** | **0.279** | 0.189 | **0.279** | 0.192 | 0.282 |
| | 192 | 0.758 | 0.736 | 0.256 | 0.325 | 0.253 | 0.319 | **0.252** | **0.317** | 0.252 | **0.317** | 0.254 | 0.319 |
| | 336 | 0.833 | 0.766 | 0.320 | 0.364 | 0.314 | 0.357 | **0.313** | **0.354** | 0.313 | **0.354** | 0.314 | 0.355 |
| | 720 | 0.946 | 0.808 | 0.424 | 0.423 | 0.414 | 0.413 | **0.412** | **0.410** | 0.413 | 0.411 | 0.413 | 0.411 |
| ECL | 96 | 0.204 | 0.318 | 0.190 | 0.309 | **0.187** | **0.304** | 0.190 | 0.308 | 0.194 | 0.312 | 0.194 | 0.311 |
| | 192 | 0.215 | 0.328 | 0.204 | 0.320 | **0.199** | **0.315** | 0.199 | 0.315 | 0.202 | 0.319 | 0.202 | 0.319 |
| | 336 | 0.227 | 0.339 | 0.216 | 0.332 | **0.212** | **0.329** | 0.212 | 0.330 | 0.216 | 0.334 | 0.217 | 0.335 |
| | 720 | 0.273 | 0.373 | 0.254 | 0.360 | **0.233** | **0.345** | 0.248 | 0.356 | 0.248 | 0.355 | 0.248 | 0.356 |

We provide additional analysis on the number of layers, and also ablations on the observation space (meaning that there is no projection into representation space by removing the embedding layer). We observe that learning deep representations lead to a significant increase in performance, and the optimal number of layers is around 2 3, before overfitting occurs.

# J  REAL-WORLD DECOMPOSED FORECASTS

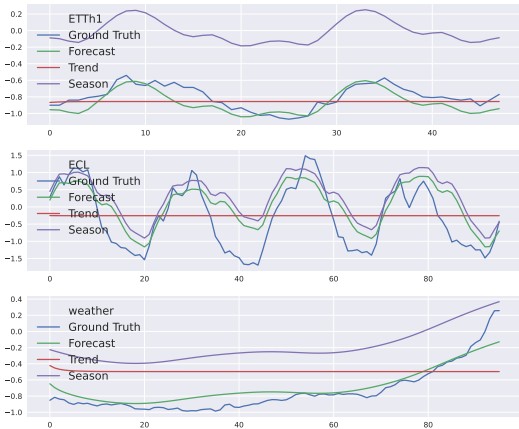

Figure 7: Visualization of decomposed forecasts from ETSformer on real world datasets, ETTh1, ECL, and Weather. Note that season is zero-centered, and trend successfully tracks the level of the time-series. Due to the long sequence forecasting setting and with a damping, the growth component is not visually obvious, but notice for the Weather dataset, the trend pattern is has a strong downward slope initially (near time step 0), and is quickly damped.

