# OpenReview forum: "ETSformer: Exponential Smoothing Transformers for Time-series Forecasting"
_ICLR.cc/2023/Conference — Submitted to ICLR 2023_

### Official Review · Reviewer_bgQv · 2022-10-23

**Confidence:** 3
**Correctness:** 2
**Technical Novelty And Significance:** 2
**Empirical Novelty And Significance:** 2
**Recommendation:** 5

**Clarity, Quality, Novelty And Reproducibility:**

Clarity on the structure of the model and how interpretability works could be further improved -- see above section for more.

**Strength And Weaknesses:**

While the model could be promising for time-series datasets where level-growth-seasonality relationships dominate, I do have several key questions:

1) Is the ETSformer only performant on time series datasets with level/trend/seasonality relationships?

A key benefit of neural forecasting models is to learn temporal dynamics in a purely data-driven way – without the users having to make any assumptions on the relationships present in the dataset. While exponential smoothing dynamics are commonly found in many time series dataset, many other types of behaviours exist (e.g. chaotic time series, event-driven behaviour, holiday seasonality, regime-switching dynamics etc.). The high-degree specialisation of the representations within the ETSformer hence raises the question is able to learn general time series relationships, or if it is confined to variants of exponential smoothing.

The model could still be useful if this were the case, but multiple comparisons would need to be made against specialised models that capture these relationships – namely simple benchmarks such as the Holts-Winter’s additive model and structural time series models, and Neural forecasters ES-RNN (winner of M4 competition) and N-Beats which also make similar decompositions.

2) How useful is the ETSformer vs common standardisations from time-series datasets?

Another key claim the paper makes is that the model does away with the need for feature engineering on the inputs. Given that some standardisation appears to have been performed on the inputs (could the authors clarify specifics?), I wonder if the performance of other transformers would be improved using standardisations more appropriate for time-series datasets – e.g. simple log or Box-Cox transforms. In addition, the ETSformer should also be compared to other networks with implicit scale handling – e.g. DeepAR which has an input scaling layer.

3) How are benchmarks calibrated?

Has hyperparameter optimisation been performed on the other comparable models, and how are hyperparameters selected otherwise?

4) Can the authors comment on how the frequency attention mechanism differs from that used by the Fedformer, and on the statistical significance of improvements vs it?

5) Is the ETSformer truly interpretable, or does it presume that the level/trend/seasonality relationships already exist?

I was a bit confused by the ESA attention plots – don’t the fixed “attention weights” pre-define the patterns to be found? In addition, are “FA weights” by definition periodic given that they are obtained by passing a limited number of frequencies through the inverse DFT? Given that none of these are learnt from the data, I am not sure that we can consider the model interpretable. Finally can the author comment on how the Real-world decomposed forecasts were obtained in the appendix? It’s not immediately clear how the components can be cleanly separated if multiple layers/stacks are present in the network.



**Summary Of The Paper:**

The paper proposes a novel deep neural network architecture for multi-horizon time series forecasting, ETSformer, via improved representation learning of level-growth-seasonality relationships commonly found in time series datasets. While the paper refers to these as attention mechanisms, it in fact models features in each layer using tweaked exponential smoothing dynamics – with seasonal components isolated using the top-k Fourier bases, and broader trends extracted using level/growth models on intermediate features.

**Summary Of The Review:**

Per the section above, I do have many key concerns regarding the paper, which would require clarification before it can be recommended for publication.

---

> ### Author Response · Authors · 2022-11-16
> **Response to Reviewer bgQv (2/2)**
>
> __Q3: How are benchmarks calibrated? Has hyperparameter optimisation been performed on the other comparable models, and how are hyperparameters selected otherwise?__
>
> A3: Benchmark results are obtained from their respective papers [1,2,3,4] in which the authors have performed extensive hyperparameter tuning as described in their papers. Some hyperparameters which have been tuned in the baselines include the lookback length, learning rate, number of layers of encoder/decoder, number of heads for multi-head attention, and model specific values such as $c$ of Auto-Correlation in Autoformer.
> For ETSformer, we tune lookback length, learning rate, and the value of $K$ for the FA mechanism. Full details can be found in Appendix D.
>
> __Q4: Can the authors comment on how the frequency attention mechanism differs from that used by FEDformer, and on the statistical significance of improvements vs it?__
>
> A4: FEDformer's Frequency Enhanced Attention replaces cross-attention by performing cross-attention in the frequency domain, relying on a complex parameterization which is parameter heavy (evidenced by the efficiency analysis in Section 3.3). On the other hand, ETSformer's Frequency Attention identifies the salient seasonal patterns in the hidden representations, and extrapolates it over the forecast horizon, leading to a lightweight but effective attention mechanism.
>
> We provide standard deviations of our benchmark results in Appendix H, Table 7. ETSformer's improvements over FEDformer are larger than 1 standard deviation, and should provide a good heuristic to establish a statistically significant improvement.
>
> __Q5. Is ETSformer truly interpretable, or does it presume that level/trend/seasonality already exist?__
>
> A5: ESA attention weights are ultimately still learned from data. While the exponential smoothing structure is predefined, $y_t = \alpha x_t + (1 - \alpha) y_{t-1}$, the exponential weight $\alpha$ is learned via gradient descent. Thus, the attention map for each dataset can be visualized. We would like to highlight that learning also takes place in the Linear layers within each layer of ETSformer, and this formulation is not too restrictive.
>
> We highlight that the chosen frequencies for the FA mechanism are not predefined, but adaptively selected via the top-K operation. The FA weights depend on the inputs which are transformed by the MLPs in each layer which are learned from the data.
>
> __Q6: How are the real-world decomposed forecasts were obtained in the appendix? It’s not immediately clear how the components can be cleanly separated if multiple layers/stacks are present in the network.__
>
> A6: The final forecast is based on Equation (1), $\hat{X}\_{t:t+H} = E\_{t:t+H} + \mathrm{Linear}(\sum\_{n=1}^N B\_{t:t+H}^{(n)} + S\_{t:t+H}^{(n)})$. Observe that the growth and seasonal components can be decomposed into $\mathrm{Linear}(\sum_{n=1}^N B_{t:t+H}^{(n)}) + \mathrm{Linear}(\sum_{n=1}^N S_{t:t+H}^{(n)})$. Thus, we can visualize each component individually.
>
> ___
>
> [1] Zhou, H., Zhang, S., Peng, J., Zhang, S., Li, J., Xiong, H., & Zhang, W. (2021, May). Informer: Beyond efficient transformer for long sequence time-series forecasting. In Proceedings of the AAAI Conference on Artificial Intelligence (Vol. 35, No. 12, pp. 11106-11115).
>
> [2] Challu, C., Olivares, K. G., Oreshkin, B. N., Garza, F., Mergenthaler, M., & Dubrawski, A. (2022). N-hits: Neural hierarchical interpolation for time series forecasting. arXiv preprint arXiv:2201.12886.
>
> [3] Wu, H., Xu, J., Wang, J., & Long, M. (2021). Autoformer: Decomposition transformers with auto-correlation for long-term series forecasting. Advances in Neural Information Processing Systems, 34, 22419-22430.
>
> [4] Zhou, T., Ma, Z., Wen, Q., Wang, X., Sun, L. &amp; Jin, R.. (2022). FEDformer: Frequency Enhanced Decomposed Transformer for Long-term Series Forecasting. <i>Proceedings of the 39th International Conference on Machine Learning</i>, in <i>Proceedings of Machine Learning Research</i> 162:27268-27286
>
> [5] Kim, T., Kim, J., Tae, Y., Park, C., Choi, J. H., & Choo, J. (2021, September). Reversible instance normalization for accurate time-series forecasting against distribution shift. In International Conference on Learning Representations.

---

> ### Author Response · Authors · 2022-11-16
> **Response to Reviewer bgQv (1/2)**
>
> Thank you Reviewer bgQv for your comments and feedback, we hope the below addresses your queries.
>
> __Q1: Is ETSformer only performant on time series datasets with level/trend/seasonality relationships?__
>
> A1: Indeed, ETSformer is endowed with the inductive biases of level/growth/seasonality and exponential smoothing as inspired by ETS methods. Other certain types of behaviors would be interesting to extend as future work such as event-driven and holiday seasonality, whereas some types of time-series such as chaotic time-series require fundamentally different solutions.
>
> That being said, we include the requested baselines of AutoETS (which considers Holt-Winter's models), N-BEATS, on the univariate benchmark.
> Results for ES-RNN can also be found in our response to all reviewers.
>
> |  Methods |     | ETSformer |           | AutoETS |       |  N-BEATS  |           |
> |:--------:|:---:|:---------:|:---------:|:-------:|:-----:|:---------:|:---------:|
> |  Metrics |     |    MSE    |    MAE    |   MSE   |  MAE  |    MSE    |    MAE    |
> |   ETTm2  |  96 | **0.080** | **0.212** |  0.794  | 0.617 |   0.082   |   0.219   |
> |          | 192 |   0.150   |   0.302   |  1.078  | 0.740 | **0.120** | **0.268** |
> |          | 336 | **0.175** | **0.334** |  1.279  | 0.822 |   0.226   |   0.370   |
> |          | 720 | **0.224** | **0.379** |  1.541  | 0.924 |   0.188   |   0.338   |
> | Exchange |  96 | **0.099** | **0.230** |  0.192  | 0.316 |   0.156   |   0.299   |
> |          | 192 | **0.223** | **0.353** |  0.355  | 0.442 |   0.669   |   0.665   |
> |          | 336 | **0.421** | **0.497** |  0.577  | 0.578 |   0.611   |   0.605   |
> |          | 720 |   1.114   | **0.807** |  1.242  | 0.865 | **1.111** |   0.860   |
>
> __Q2: What are the pre-processing techniques used by ETSformer, and how does this compare to the baselines?__
>
> A2: In our work, the only data pre-processing on data inputs that we perform is standardisation, meaning that we calculate the mean and standard deviation from the training set, and perform $(x-mean)/std$. We highlight that this is a standard pre-processing technique for the LSTF benchmark as established by prior work [1,2,3,4], that we follow, meaning all the baseline Transformers also use this pre-processing.
>
> Performing more advanced data pre-processing techniques is not a standard in this line of research, one reason being deep learning methods are in fact attempting to learn representations or some transformation to the data. Another reason is that more advanced normalization/standardization methods is an orthogonal research direction, where plug-and-play normalization modules which can be attached to any method have been introduced [5].
>
> Here are some comparisions to DeepAR from the univariate benchmark, note that DeepAR was not proposed for long sequence time-series forecasting, and suffers from further drawbacks including high computational cost due to unrolling across long horizons
>
> |  Methods |     | ETSformer |           | DeepAR |       |
> |:--------:|:---:|:---------:|:---------:|:------:|:-----:|
> |  Metrics |     |    MSE    |    MAE    |   MSE  |  MAE  |
> |   ETTm2  |  96 | **0.080** | **0.212** |  0.099 | 0.237 |
> |          | 192 | **0.150** | **0.302** |  0.154 | 0.310 |
> |          | 336 | **0.175** | **0.334** |  0.277 | 0.428 |
> |          | 720 | **0.224** | **0.379** |  0.332 | 0.468 |
> | Exchange |  96 | **0.099** | **0.230** |  0.417 | 0.515 |
> |          | 192 | **0.223** | **0.353** |  0.813 | 0.735 |
> |          | 336 | **0.421** | **0.497** |  1.331 | 0.962 |
> |          | 720 | **1.114** | **0.807** |  1.890 | 1.181 |

---

> ### Author Response · Authors · 2022-12-01
> **Further Discussions**
>
> Dear Reviewer bgQv,
>
> We hope that you have had the time to go through our response and revised manuscript. This is a gentle reminder and request, if our response has been satisfactory, that you consider raising your recommendation. If not, we look forward to further discussions with you.
>
> Best regards,
>
> Authors

---

> > ### Author Response · Authors · 2022-12-05
> > **Gentle Reminder**
> >
> > Dear Reviewer bgQv,
> >
> > Hope you are well! As the discussion period is coming to an end soon, we are looking forward to any feedback you have to our response despite your busy schedule. We will highly appreciate it if you let us know whether your previous concerns have been adequately addressed. Thank you once again.
> >
> > We have done our utmost to address the concerns as you suggested:
> >
> > 1. We have __highlighted the differences__ between ETSformer, and existing work such as FEDformer, and ES-RNN.
> > 2. We have __included additional empirical comparisons__ between the important baseline of ES-RNN.
> > 3. We have __addressed the reviewer's queries__, regarding benchmark hyperparameter tuning, interpretability of ETSformer, etc.
> >
> > Thank you again for your dedication to reviewing our paper and we are looking forward to your feedback.

---

> ### Author Response · Authors · 2022-12-07
> **Gentle reminder for further comments**
>
> Dear Reviewer bgQv,
>
> Hope everything is going well. As the discussion period is coming to an end very soon, we would like to send a gentle reminder, and to let you know that we look forward to hearing any further updates and thoughts you may have. Once again, thank you for your efforts and service.

---

### Official Review · Reviewer_7t86 · 2022-10-23

**Confidence:** 4
**Correctness:** 3
**Technical Novelty And Significance:** 3
**Empirical Novelty And Significance:** 3
**Recommendation:** 6

**Clarity, Quality, Novelty And Reproducibility:**

This paper is well written. The idea is novel and the contributions are clear. The authors have also included the code for reproducing the results.

**Strength And Weaknesses:**

Strength:
1. The novelty of this paper is clear and well-motivated. The exponential smoothing method is widely used in time series literature and the underlying supports of ETSformer, Holt-Winter's method, and its modifications are also theoretically sound.
2. To obtain the attention matrix, this paper uses an efficient $\mathcal{A}_{ES}$ that computes the attention in $\mathcal{O}(L^2)$ complexity, which reduces the computational time compared with baselines.
3. The empirical evaluation and ablation study are thorough and sound, providing good support for the proposed method.

Weakness:
1. Exponential smoothing also has some limitations in time series forecasting, such as (1) the forecasts it generates will be behind, (2) difficult to account for dynamic changes in the real world, and (3) constantly requires updating to respond to new information. The authors need to discuss how the proposed ETSformer can overcome these drawbacks compared with classical methods.
2. I am not sure what is the benefit of having $N$ layers/stacks together in the encoder/decoder. It seems that they are not time stamps. To better explain the intuition, it is good to have an e.g., layer-wise analysis to visualize/quantitatively show the difference of the information captured at each layer.
3. Although ETSformer is more efficient, it appears the improvement over FEDformer is not much.

Minor Questions:
1. Are the empirical evaluations single-step or multi-step forecasts?
2. In the first paragraph of section 3, what does the "dimension" mean? Is this a temporal dimension or a feature dimension?

**Summary Of The Paper:**

This paper proposed a transformer architecture called ETSformer that leverage the advantage of the exponential smoothing method for more accurate and interpretable forecasts. The core contributions are the development of exponential smoothing attention (ESA) and frequency attention (FA) mechanisms that model the growth and seasonality information of time series data. These methods build connections with classical time series models and perform forecasts for each individual component of time series data, leading to better point forecast accuracy and interpretability. The overall transformer architecture is clear and intuitive. Empirical results have shown improved forecasting accuracy than other transformer-based forecasting baselines, and have shown better computational efficiency and interpretability.

**Summary Of The Review:**

Overall, this paper provides a novel modification for the transformer architecture that can improve both the accuracy and interpretabilty of time series forecasts. The proposed method is proven to work well in multiple real datasets. As an improvement, the authors can better clarify how the proposed method can overcome classical exponential smoothing drawbacks and a more intuitive explanation of the model architectures.

**Update after rebuttal**

The authors have addressed my concerns and have demonstrated them with additional experiments. Based on the response and comments from other reviewers, I tend to keep my current evaluation.

---

> ### Author Response · Authors · 2022-11-16
> **Response to Reviewer 7t86 (2/2)**
>
> __Q3: How does ETSformer improve over FEDformer apart from efficiency?__
>
> A3: Apart from efficiency, ETSformer demonstrates more disentangled seasonal-trend forecasts as observed in Figure 1. Furthermore, ETSformer is more interpretable as can be seen by the visualized attention weights in Figure 4, demonstrating the learned seasonal dependencies present in the dataset. FEDformer does not have such interpretable capabilities due to its complex Frequency Enhanced Attention mechanism.
>
> __Q4: Are the empirical evaluations single-step or multi-step forecasts?__
>
> A4: The evaluations are made over long horizon lengths, indicated as 96/192/336/720 in the results tables. Depending on the method, forecasts can be made in a single forward pass (direct multi-step), or can be made iteratively (iterative multi-step).
>
> __Q5: In the first paragraph of section 3, what does the "dimension" mean? Is this a temporal dimension or a feature dimension?__
>
> A5: Dimension refers to the feature dimension. We highlight that this is a standard experimental setting as established by previous work [1,2].
>
> ---
> [1] Wu, H., Xu, J., Wang, J., & Long, M. (2021). Autoformer: Decomposition transformers with auto-correlation for long-term series forecasting. Advances in Neural Information Processing Systems, 34, 22419-22430.
>
> [2] Zhou, T., Ma, Z., Wen, Q., Wang, X., Sun, L. &amp; Jin, R.. (2022). FEDformer: Frequency Enhanced Decomposed Transformer for Long-term Series Forecasting. Proceedings of the 39th International Conference on Machine Learning, in Proceedings of Machine Learning Research 162:27268-27286

---

> ### Author Response · Authors · 2022-11-16
> **Response to Reviewer 7t86 (1/2)**
>
> Thank you Reviewer 7t86 for the comments and feedback, we hope the below addresses your queries.
>
> __Q1: Classical exponential smoothing methods face the following limitations: (1) the forecasts it generates will be behind, (2) difficult to account for dynamic changes in the real world, and (3) constantly requires updating to respond to new information. Please discuss how the proposed ETSformer can overcome these drawbacks.__
>
> A1:
>
> (1) This criticism typically holds only for simple exponential smoothing methods. In contrast, ETSformer learns hierarchical level-growth-season representations with it's Transformer-based architecture. It also leverages its growth damping and seasonal extrapolation module to overcome this limitation. Furthermore, with it's deep formulation which has more learnable parameters within the MLPs, ETSformer is able to perform forecasts rather than simply performing a smoothing operation.
>
> (2) Indeed, dynamic changes in the real world are hard to account for. ETSformer accounts for certain types of dynamic changes such as seasonality, and growth factors that are predictable.
> ETSformer currently does not account for certain factors such as holidays (which is an interesting future direction which can be incorporated), and irregular shocks (which is more difficult to handle, and may require more information from the world rather than just the available time-series).
>
> (3) Classical ETS methods follow a recurrent formulation (similar to RNNs) and thus require this constant updating to respond to new information. By basing ETSformer on a Transformer-based methodology, we assume that the forecast horizon is only dependent on the lookback window and bypass the recurrent formulation.
>
> __Q2: What is the benefit of having $N$ layers/stacks?__
>
> A2: Deeper models typically lead to improved performance due to larger model capacity, and being able to learn better hierarchical representations. This improvement typically reverses at a certain number of layers due to overfitting as the model becomes overparameterized. We provide additional analysis on the number of layers, and also analysis on the observation space (Obs Space refers to a variation of ETSformer which removes the embedding projection layer, and performs operations in observation space). We observe that learning deep representations lead to a significant increase in performance, and the optimal number of layers is around 2~3, before overfitting occurs.
>
> | Num   Layers |     | Obs. Space |       | Layers=1 |       |  Layers=2 |           |  Layers=3 |           |  Layers=4 |           | Layers=5 |       |
> |:------------:|:---:|:----------:|:-----:|:--------:|:-----:|:---------:|:---------:|:---------:|:---------:|:---------:|:---------:|:--------:|:-----:|
> |    Metrics   |     |     MSE    |  MAE  |    MSE   |  MAE  |    MSE    |    MAE    |    MSE    |    MAE    |    MSE    |    MAE    |    MSE   |  MAE  |
> |     ETTm2    |  96 |    0.685   | 0.705 |   0.190  | 0.284 |   0.189   |   0.280   | **0.188** | **0.279** |   0.189   | **0.279** |   0.192  | 0.282 |
> |              | 192 |    0.758   | 0.736 |   0.256  | 0.325 |   0.253   |   0.319   | **0.252** | **0.317** | **0.252** | **0.317** |   0.254  | 0.319 |
> |              | 336 |    0.833   | 0.766 |   0.320  | 0.364 |   0.314   |   0.357   | **0.313** | **0.354** | **0.313** | **0.354** |   0.314  | 0.355 |
> |              | 720 |    0.946   | 0.808 |   0.424  | 0.423 |   0.414   |   0.413   | **0.412** | **0.410** |   0.413   |   0.411   |   0.413  | 0.411 |
> |      ECL     |  96 |    0.204   | 0.318 |   0.190  | 0.309 | **0.187** | **0.304** |   0.190   |   0.308   |   0.194   |   0.312   |   0.194  | 0.311 |
> |              | 192 |    0.215   | 0.328 |   0.204  | 0.320 | **0.199** | **0.315** |   0.199   |   0.315   |   0.202   |   0.319   |   0.202  | 0.319 |
> |              | 336 |    0.227   | 0.339 |   0.216  | 0.332 | **0.212** | **0.329** |   0.212   |   0.330   |   0.216   |   0.334   |   0.217  | 0.335 |
> |              | 720 |    0.273   | 0.373 |   0.254  | 0.360 | **0.233** | **0.345** |   0.248   |   0.356   |   0.248   |   0.355   |   0.248  | 0.356 |

---

> ### Author Response · Authors · 2022-12-01
> **Further Discussions**
>
> Dear Reviewer 7t86,
>
> We hope that you have had the time to go through our response and revised manuscript. This is a gentle reminder and request, if our response has been satisfactory, that you consider raising your recommendation. If not, we look forward to further discussions with you.
>
> Best regards,
>
> Authors

---

> > ### Author Response · Authors · 2022-12-05
> > **Gentle Reminder**
> >
> > Dear Reviewer 7t86,
> >
> > Hope you are well! As the discussion period is coming to an end soon, we are looking forward to any feedback you have to our response despite your busy schedule. We will highly appreciate it if you let us know whether your previous concerns have been adequately addressed. Thank you once again.
> >
> > We have done our utmost to address the concerns as you suggested:
> >
> > 1. We have __addressed the reviewer's queries__ regarding the limitations of ETS, and other minor questions.
> > 2. We have included __additional empirical analysis__, performing an ablation study on the number of layers/stacks.
> >
> > Thank you again for your dedication to reviewing our paper and we are looking forward to your feedback.

---

> ### Author Response · Authors · 2022-12-07
> **Gentle reminder for further comments**
>
> Dear Reviewer 7t86,
>
> Hope everything is going well. As the discussion period is coming to an end very soon, we would like to send a gentle reminder, and to let you know that we look forward to hearing any further updates and thoughts you may have. Once again, thank you for your efforts and service.

---

### Official Review · Reviewer_9rNK · 2022-10-24

**Confidence:** 4
**Correctness:** 3
**Technical Novelty And Significance:** 2
**Empirical Novelty And Significance:** 2
**Recommendation:** 5

**Clarity, Quality, Novelty And Reproducibility:**

The paper is written clearly. The method novelty is marginal. It looks reproducible.

**Strength And Weaknesses:**

Strengths:
1. It's novel to combine traditional exponential smoothing method with Transformer based model. Also it replaces the self-attention in vanilla Transformer with their designed frequency attention mechanism.
2. Exponential Smoothing Attention and Frequency Attention Mechanism are derived in detail.
3. Qualitative and quantitative experiments demonstrate the efficacy of proposed method.

Weaknesses:
1. Previously exponential smoothing has already been integrated with LSTM. The paper doesn't show the comparison of the proposed method with the integration with LSTM.
2. It's not clear that whether this special Exponential Smoothing Attention and Frequency Attention Mechanism could be integrated with other Transformer based model, e.g. Autoformer, FedFormer and etc.
3. The comparison is out-of-dated. It doesn't compare with FedFormer and NHITS.

**Summary Of The Paper:**

This paper proposes a novel exponential smoothing at- tention and frequency attention to replace the self-attention mechanism in vanilla Transformers.  They also redesign the Transformer architecture with modular decomposition blocks such that it can learn to decompose the time-series data into interpretable time- series components such as level, growth and seasonality. Experimental results show the efficacy of proposed method.

**Summary Of The Review:**

This paper proposes a novel exponential smoothing at- tention and frequency attention to replace the self-attention mechanism in vanilla Transformers.  They also redesign the Transformer architecture with modular decomposition blocks such that it can learn to decompose the time-series data into interpretable time- series components such as level, growth and seasonality. Experimental results show the efficacy of proposed method.

However,  It's not clear that whether this special Exponential Smoothing Attention and Frequency Attention Mechanism could be integrated with other Transformer based model, e.g. Autoformer, FedFormer and etc. Also the integration of Exponential smoothing with some backbone neural network is not novel. (previously Uber did the integration with LSTM). It doesn't show the experimental results difference.

---

> ### Author Response · Authors · 2022-11-16
> **Response to Reviewer 9rNK**
>
> Thank you Reviewer 9rNK for the comments and feedback. We hope that the below answers your queries and clears any doubts you may have had.
>
> __Q1: The paper does not include comparisons with existing work which combines exponential smoothing with LSTM (the ES-RNN model)?__
>
> A1: Thank you for raising this point up, we have included ES-RNN as a new baseline model, and further elaborated upon the differences of our work with theirs. Please see the response to all reviewers, and our revised manuscript.
>
> __Q2: Are the special Exponential Smoothing Attention and Frequency Attention Mechansims able to be integrated with other Transformer based models (e.g. Autoformer, FEDformer, etc.)?__
>
> A2: Our proposed Exponential Smoothing Attention and Frequency Attention mechanisms are designed to work hand-in-hand with our novel level-growth-season decomposition methodology. On the other hand, Autoformer and FEDformer follow the same decomposition strategy while swapping out different seasonality extraction mechanisms. Thus, our proposed ESA and FA mechanisms are not designed to be integrated with these other Transformer based models. That being said, we reiterate that our proposed decomposition architecture along with ESA/FA mechanisms have the added benefit of being more interpretable, as well as more efficient compared to that presented in Autoformer and FEDformer.
>
> __Q3: The comparison is outdated. It doesn't compare with FEDFormer and NHITS.__
>
> A3: One of our main comparisons is in fact with FEDformer, where we not only show that ETSformer achieves comparable performance with FEDformer (outperforming FEDformer in certain settings), but ETSformer is also beats FEDformer in two other aspects: interpretability (Section 3.3), and efficiency (Section 3.4).
> With regards to NHITS, they are concurrent work which is only found on arXiv and currently not published. That being said, we gladly include it in our related work section for a richer literature review in our revised manuscript.

---

> ### Author Response · Authors · 2022-12-01
> **Further Discussions**
>
> Dear Reviewer 9rNK,
>
> We hope that you have had the time to go through our response and revised manuscript. This is a gentle reminder and request, if our response has been satisfactory, that you consider raising your recommendation. If not, we look forward to further discussions with you.
>
> Best regards,
>
> Authors

---

> > ### Author Response · Authors · 2022-12-05
> > **Gentle Reminder**
> >
> > Dear Reviewer 9rNK,
> >
> > Hope you are well! As the discussion period is coming to an end soon, we are looking forward to any feedback you have to our response despite your busy schedule. We will highly appreciate it if you let us know whether your previous concerns have been adequately addressed. Thank you once again.
> >
> > We have done our utmost to address the concerns as you suggested:
> >
> > 1. We have included an __additional empirical comparison__ with existing work combines ETS with neural networks (ES-RNN), as well as highlight the differences, novelty, and strengths of ETSformer over ES-RNN.
> > 2. We have addressed the reviewer's queries regarding our proposed attention mechanisms.
> > 3. We have clarified that ETSformer compares directly in terms of performance, interpretability, and efficiency with competing baseline FEDformer.
> >
> > Thank you again for your dedication to reviewing our paper and we are looking forward to your feedback.

---

> ### Author Response · Authors · 2022-12-07
> **Gentle reminder for further comments**
>
> Dear Reviewer 9rNK,
>
> Hope everything is going well. As the discussion period is coming to an end very soon, we would like to send a gentle reminder, and to let you know that we look forward to hearing any further updates and thoughts you may have. Once again, thank you for your efforts and service.

---

### Official Review · Reviewer_cG8G · 2022-11-05

**Confidence:** 4
**Correctness:** 2
**Technical Novelty And Significance:** 2
**Empirical Novelty And Significance:** 2
**Recommendation:** 3

**Clarity, Quality, Novelty And Reproducibility:**

The overall clarify is OK. However, the technical quality and novelty is limited. Reproducibility is OK since code is available.


**Details Of Ethics Concerns:**

No.

**Strength And Weaknesses:**

Strength:

1. This paper is well-written and organized.
2. Several transformer-based baselines have been used in empirical studies.

Weaknesses:

1. Combining exponential smoothing with neural networks to perform time series forecasting is not a new idea, for example,

[1] A hybrid method of Exponential Smoothing and Recurrent Neural Networks for time series forecasting International Journal of Forecasting 2019

[2] A Hybrid Residual Dilated LSTM and Exponential Smoothing Model for Midterm Electric Load Forecasting. IEEE TNNLS 2021

[3] Time‑series analysis with smoothed Convolutional Neural Network

Especially In [2], the authors have thoroughly assessed various approaches (including LSTM, GNNs)+ ETS for time series forecasting. However, none of them were mentioned or compared in this paper. Also, applying ETS to the transformer seems straightforward to me and thus the technical novelty here is limited.

2. FEDformer has already shown that properly considering frequency-based attention in the transformer is useful for long-range forecasting. In the paper, I did not observe how the proposed FA is superior to FEDformer.

3. Overall, I feel this paper is a little bit add-hoc, i.e., combining the benefit of ETS (which has been validated in earlier works) and FA (similar ideas have been validated in FEDformer).  It is not clear which component contributes more to the performance boost.

4. The experiment results do not show the obvious superior performance of the proposed ETSformer over FEDformer and other baselines.

5. The discussion of the results in Table 1 is limited, especially for the cases when ETSformer cannot outperform baselines.


**Summary Of The Paper:**

This paper presents exponential smoothing transformers for long-range time series forecasting. The key idea is to leverage a level-growth-seasonality decomposed transformer architecture and employ both exponential smoothing attention and frequency attention to reduce computational complexity. The experiment results showed the effectiveness of the proposed method.

**Summary Of The Review:**

Overall, I have concerns over the novelty, related works, and experiment results.

---

> ### Author Response · Authors · 2022-11-16
> **Response to Reviewer cG8G (2/2)**
>
> __Q3: The paper feels a little bit add-hoc, combining existing exponential smoothing ideas and frequency based attention mechanisms. Which component contributes more to performance boost?__
>
> A3: We highlight that our proposed approach is not an ad-hoc combination of exponential smoothing and frequency based attention mechanism. Rather, our design methodology is as follows:
>
> 1. From the starting point of level-growth-season decomposition from ETS methods, we designed a novel method to fuse this decomposition into the Transformer layers.
> 2. Then, we incorporated exponential smoothing inductive biases for the level and growth components.
> 3. Finally, to overcome the downside of requiring predefined season periods in classical ETS methods, we designed this novel Frequency Attention mechanism which can automatically extract the salient periodicity from hidden representations.
>
> For a better understanding of which component contributes to performance boost, please take a look at our extensive ablations in Section 3.2, Tables 2 and 3.
>
> __Q4: Experiment results do not show obvious superior performance over FEDformer baseline, and discussion of results in Table 1 is limited.__
>
> A4: Indeed, FEDformer was concurrent work developed during the same period. In terms of performance based on MSE/MAE, we achieve similar results. However, we would like to highlight our following contributions and benefits over FEDformer:
>
> 1. ETSformer introduces a novel decomposition scheme, whereas FEDformer follows the decomposition scheme from older work [3].
> 2. ETSformer introduces novel attention mechanisms Exponential Smoothing Attention and Frequency Attention which are distinct from FEDformer's Frequency Enhanced Attention.
> 3. Based on the above proposals, ETSformer achieves superior interpretability (Section 3.3), and efficiency (Section 3.4).
>
> ---
>
> [1] Smyl, S. (2020). A hybrid method of exponential smoothing and recurrent neural networks for time series forecasting. International Journal of Forecasting, 36(1), 75-85.
>
> [2] Dudek, G., Pełka, P., & Smyl, S. (2021). A hybrid residual dilated LSTM and exponential smoothing model for midterm electric load forecasting. IEEE Transactions on Neural Networks and Learning Systems.
>
> [3] Wu, H., Xu, J., Wang, J., & Long, M. (2021). Autoformer: Decomposition transformers with auto-correlation for long-term series forecasting. Advances in Neural Information Processing Systems, 34, 22419-22430.

---

> ### Author Response · Authors · 2022-11-16
> **Response to Reviewer cG8G (1/2)**
>
> Thank you Reviewer cG8G for the comments and questions. We have addressed your queries and concerns in the specific questions below. Overall, we note that your major concern is regarding the novelty of our work -- we hope that our response to Q1 will make clear the novelty of our work, and how it is different from any existing work.
>
> __Q1: What is the difference between ETSformer and existing works which have already combined exponential smoothing with neural networks [1]?__
>
> A1: While combining exponential smoothing with neural networks has been explored by the ES-RNN model [1,2], we highlight that our proposed method is significantly different.
> * ES-RNN leverages exponential smoothing for pre and post processing of the time-series. For each time-series, they learn multiplicative seasonal and level exponential weights. Pre-processing is done by removing (dividing) this level and seasonal component, before feeding in the processed data into a neural network (RNN, LSTM, etc.). Finally, post-processing is performed by multiplying the season and level values back as forecasts.
> * On the other hand, we leverage the power of a deep architecture to learn level-growth-season __representations__, embedding the notions of level-growth-seasonal decomposition into the Transformer model layers as inductive biases for time-series data. Note that we do not simply replace the vanilla attention with an exponential smoothing weighting, but actually introduce a novel and non-trivial level-growth-seasonal decomposition scheme along with the ESA and FA mechanisms.
>
> Thus, we reiterate, that ___our work is not simply taking ES-RNN to replace the RNN/LSTM/etc. with a Transformer.___
> Next, we highlight the novelty of our decomposition architecture, ESA, and FA, which are new attention mechanisms designed to suit the characteristics of time-series:
> * While our decomposition architecture was inspired by ETS methods, the design is completely novel as it is embedded in the framework of Transformers, leading to decomposed level-growth-season representations. By performing decomposition in representation space, ETSformer is able to learn more complex relationships between the different components. This is evidenced by our superior interpretability compared to Autoformer and FEDformer, in Section 3.3.
> * ESA mechanism embeds the idea of exponential weighting into attention weights. The inductive bias that weights should decay exponentially is critical in time-series data to model growth/trend. Using this to replace vanilla attention reduces the number of learnable parameters (increase efficiency), and also reduces overfitting to noise.
> * FA mechanism works hand in hand with the decomposition architecture to automatically identify seasonality in the data, unlike classical ETS methods which require prior knowledge on the periodicity. We further highlight the details of FA in the next question.
>
> Therefore, we respectfully disagree with the characterization that applying ETS to Transformer is straightforward with limited technical novelty.
> At the same time, we also thank the reviewer for the reminder of the very relevant baseline of ES-RNN, which we include as a new baseline and add to related work. Please do take a look at our response to all reviewers and the revised manuscript.
>
> __Q2: How does the Frequency Attention mechanism compare to FEDformer's Frequency Enhanced Attention?__
>
> A2: We first highlight that our Frequency Attention (FA) mechanism is a novel contribution, and operates very differently from FEDformer's Frequency Enhanced Attention (FEA). ETSformer's FA extracts the salient periodicities via frequency-domain analysis, and extrapolates these periodicities over the forecast horizon. On the other hand, FEDformer's FEA relies on learning a complex and parameter heavy transformation in the frequency domain.
>
> Next, we highlight several aspects in which our FA mechanism performs more favorably on.
>
> 1. Interpretability. As demonstrated in Figures 1, the contributions from ETSformer produces a more disentangled seasonal-trend decomposition as compared to FEDformer. Furthermore in Figure 4, the FA mechanism has better interpretability as it exhibits a clear periodical pattern which can be extracted. FEDformer's FEA provides no such capabilities due to its complex formulation.
> 2. Efficiency. As demonstrated in Figure 6, FEDformer's FEA leads to subpar runtime and memory efficiency as compared to ETSformer.

---

> ### Author Response · Authors · 2022-12-01
> **Further Discussions**
>
> Dear Reviewer cG8G,
>
> We hope that you have had the time to go through our response and revised manuscript. This is a gentle reminder and request, if our response has been satisfactory, that you consider raising your recommendation. If not, we look forward to further discussions with you.
>
> Best regards,
>
> Authors

---

> > ### Author Response · Authors · 2022-12-05
> > **Gentle Reminder**
> >
> > Dear Reviewer cG8G,
> >
> > Hope you are well! As the discussion period is coming to an end soon, we are looking forward to any feedback you have to our response despite your busy schedule. We will highly appreciate it if you let us know whether your previous concerns have been adequately addressed. Thank you once again.
> >
> > We have done our utmost to address the concerns as you suggested:
> >
> > 1. We have __clarified the possible misunderstanding__ that ETSformer is an ad-hoc combination of ETS and frequency-based attention mechanisms.
> > 2. We have __demonstrated that ETSformer is a novel contribution__, different from existing ES-RNN and FEDformer.
> > 3. We __conducted a new empirical comparison__ with important baseline of ES-RNN as raised by the reviewer.
> >
> > Thank you again for your dedication to reviewing our paper and we are looking forward to your feedback.

---

> ### Author Response · Authors · 2022-12-07
> **Gentle reminder for further comments**
>
> Dear Reviewer cG8G,
>
> Hope everything is going well. As the discussion period is coming to an end very soon, we would like to send a gentle reminder, and to let you know that we look forward to hearing any further updates and thoughts you may have. Once again, thank you for your efforts and service.

---

### Author Response · Authors · 2022-11-16
**Response to all reviewers (1/2)**

Thank you to all reviewers for taking the time and effort to read through our work and giving comments and feedback.

Overall, reviewers had positive opinions of our paper, that the __paper is well-written and organized__ (cG8G).
Regarding the methodology, reviewers thought that __the novelty of this paper is clear and well-motivated__ (7t86), that it is __novel to combine traditional exponential smoothing method with Transformer based model__ (9rNK), and that the __Exponential Smoothing Attention and Frequency Attention Mechanism are derived in detail__ (9rNK).
Finally, the reviewers also thought that __the empirical evaluation and ablation study are thorough and sound, providing good support for the proposed method__ (7t86).

At the same time, reviewers also raised several concerns and constructive criticisms which we have exerted our best efforts in addressing. We have made highlighted in blue any changes made in our revised manuscript, and also summarized them below:

* Updated experiments section to include ES-RNN baseline.
* Updated related works section to include ES-RNN and N-HiTS (concurrent work).
* Included Appendix I Layer Analysis, an additional sensitivity analysis study on number of layers/stacks of ETSformer.

Finally, we cover some key issues that was common among the feedback from many reviewers. We look forward to engaging the reviewers and are committed to address any further concerns.

---

> ### Author Response · Authors · 2022-11-19
> **Response to all reviewers (2/2)**
>
> __Q: The ES-RNN model [1] is an important baseline, since it is prior work which has combined exponential smoothing with neural networks. What are the key differences between ETSformer and ES-RNN, and how do their performance compare?__
>
> 1. ETSformer is a completely different approach from existing ES-RNN in combining ETS with neural networks:
>     * ES-RNN: use ETS as a simple pre and post processing method, to remove seasonality and level, then, allowing the neural network to model trend.
>     * ETSformer: completely embed the ideas of ETS into Transformer framework:
>         * Design a novel Transformer-based architecture with level-growth-season decomposition concept,
>         * Introduce novel attention mechanisms, ESA and FA mechanisms, to replace the original self-attention to handle trend and seasonality separately.
> 2. ES-RNN is designed to specifically handle positive time-series (only allows for positive values).
> 3. ES-RNN requires prior knowledge of periodicity.
> 4. ES-RNN was not designed for long sequence time-series forecasting, leading to extremely high computation costs. This is evidenced by the tuning of ES-RNN on the ETTm2 dataset over 4 lookback windows and 3 learning rates taking ~24 gpu-days, whereas ETSformer took only several gpu-hours.
>
> That being said, we have done our best to adjust ES-RNN in order to evaluate on the popular Long Sequence Time-series Forecasting benchmark. Specifically, we took the following steps to adapt the ES-RNN model for the LSTF benchmark: (1) shift time-series values such that they become positive, (2) predefine periodicity for ETS. Due to limitations on time and computation resources, we currently are still running experiments on certain settings. We seek your understanding for the delay, and are committed to adding all the results.
>
> |  Methods |     | ETSformer |           | ES-RNN |       |
> |:--------:|:---:|:---------:|:---------:|:------:|:-----:|
> |  Metrics |     |    MSE    |    MAE    |   MSE  |  MAE  |
> |   ETTm2  |  96 | **0.189** | **0.280** |  0.204 | 0.323 |
> |          | 192 | **0.253** | **0.319** |  0.351 | 0.405 |
> |          | 336 | **0.314** | **0.357** |  0.476 | 0.485 |
> |          | 720 | **0.414** | **0.413** |  0.623 | 0.561 |
> |    ECL   |  96 | **0.187** | **0.304** |  0.922 | 0.666 |
> |          | 192 | **0.199** | **0.315** |  0.499 | 0.479 |
> |          | 336 | **0.212** | **0.329** |  0.760 | 0.570 |
> |          | 720 | **0.233** | **0.345** |    -   |   -   |
> | Exchange |  96 | **0.085** | **0.204** |  0.096 | 0.221 |
> |          | 192 | **0.182** | **0.303** |  0.214 | 0.360 |
> |          | 336 | **0.348** | **0.428** |  0.469 | 0.537 |
> |          | 720 | **1.025** | **0.774** |  1.997 | 1.143 |
> |  Traffic |  96 | **0.607** | **0.392** |  1.315 | 0.546 |
> |          | 192 | **0.621** | **0.399** |  0.727 | 0.373 |
> |          | 336 | **0.622** | **0.396** |    -   |   -   |
> |          | 720 | **0.632** | **0.396** |    -   |   -   |
> |  Weather |  96 | **0.197** | **0.281** |  0.585 | 0.507 |
> |          | 192 | **0.237** | **0.312** |  0.381 | 0.397 |
> |          | 336 | **0.298** | **0.353** |  0.628 | 0.533 |
> |          | 720 | **0.352** | **0.388** |  0.711 | 0.545 |
> |    ILI   |  24 | **2.527** | **1.020** |  5.393 | 1.561 |
> |          |  36 | **2.615** | **1.007** |  6.478 | 1.751 |
> |          |  48 | **2.359** | **0.972** |  7.160 | 1.963 |
> |          |  60 | **2.487** | **1.016** |  5.801 | 1.711 |
>
> We observe that while ES-RNN obtains better results compared to simple LSTM and LSTNet methods, and even Informer, ETSformer still outperforms ES-RNN. We also see ETSformer increasingly outperform ES-RNN as horizon length grows, verifying ETSformer's superior design for Long Sequence Time-series Forecasting.
>
> ---
>
> [1] Smyl, S. (2020). A hybrid method of exponential smoothing and recurrent neural networks for time series forecasting. International Journal of Forecasting, 36(1), 75-85.

---

### Author Response · Authors · 2022-11-25
**Gentle reminder to reviewers**

Dear Reviewers,

We hope that our responses and revised manuscript has resolved any doubts you may have had with our work. This is a gentle reminder and request, if our updates have been satisfactory, that you could consider raising your recommendation. If not, please do let us know what are the remaining issues, and we are happy to have further discussions.

Best regards,

Authors

---

### Decision · Program_Chairs · 2023-01-20

**Decision:**

Reject

**Justification For Why Not Higher Score:**

This paper has potential to cross the acceptance bar after substantial revision. Currently it is not up to the standard of ICLR.


**Justification For Why Not Lower Score:**

N/A


**Metareview: Summary, Strengths And Weaknesses:**

It is useful to develop time series forecasting models that are more interpretable or decomposable, especially when the models are deep learning models. It seems novel to combine the traditional exponential smoothing method with transformer-based models in particular, although exponential smoothing was combined with other neural network models before. This is a natural next step which has already happened to many other deep learning models. The attention mechanism of the proposed model is more efficient than other baselines. However, the technical novelty of this method is not very significant. Moreover, the proposed model can only handle level, trend and seasonality behaviors. When other behaviors co-exist with these (which is not uncommon in practice), can the proposed method still work well for these behaviors? Unless there is theoretical justification, extensive experimentation will be needed to demonstrate the robustness of the model even when other behaviors exist. While we generally agree that this work has merits, it is hard to champion acceptance of the paper, even in its revised form. We hope the authors will make substantial revision of their paper for future resubmission.